# Momentum flux characteristics of vertically propagating gravity waves

Prosper K. Nyassor[1], Cristiano M. Wrasse[1], Igo Paulino[2], Erdal Yiğit[3], Vera Y. Tsali-Brown[4],
Ricardo A. Buriti[2], Cosme A. O. B. Figueiredo[2], Gabriel A. Giongo[1], Fábio Egito[2],
Oluwasegun M. Adebayo[5], Hisao Takahashi[1], and Delano Gobbi[1]

[1]Space Weather Division, National Institute of Space Research (INPE), São José dos Campos, Brazil
[2]Academic Unit of Physics, Federal University of Campina Grande (UFCG), Campina Grande, Brazil
[3]Department of Physics and Astronomy, George Mason University, Fairfax, VA, USA
[4]Instituto de Pesquisa e Desenvolvimento (IP&D), Universidade do Vale do Paraíba (UNIVAP),
São José dos Campos, Brazil
[5]Division of Heliophysics, Planetary Science and Aeronomy, INPE, São José dos Campos, Brazil CE1

**Correspondence:** Prosper K. Nyassor (prosper.nyassor@inpe.br)

**Abstract.** CE2 Momentum flux and propagation dynamics of two vertically propagating atmospheric gravity waves (GWs) are studied using observations at São João do Cariri (7.40° S, 36.31° W), Brazil, from co-located photometer, all-sky imager, and meteor radar instruments. Time series of the atomic oxygen green line (OI 557.7 nm), molecular oxygen ($O_2$ (0–1)), sodium D-line (NaD), and hydroxyl (OH (6–2)) airglow intensity variations measured by the photometer were used to investigate the vertical characteristics and vertical phase progression of the GWs with similar ($\pm 10\%$ of the error margin) or nearly the same ($\pm 5\%$ of the error margin) CE3 period across these emission layers. The horizontal parameters of the same GWs were determined from the OH airglow images, whereas the intrinsic parameters of the horizontal and vertical components of the GWs were estimated with the aid of the observed winds. Using the phase of the GWs at each emission layer, the characteristics of the phase progression exhibited near-vertical propagation under a duct background propagation condition. This indicates that the duct contributes significantly to the observed near-vertical phase propagation. The GW momentum flux and potential energy were estimated using the rotational temperatures of OH and $O_2$, revealing CE4 that the time series of momentum fluxes and potential energies are higher in the $O_2$ emission band than in the OH band, indicating a transfer of momentum and energy across OH to the $O_2$ altitude. These results reveal the effect of a duct on vertically propagating GWs and the associated momentum flux and potential energy transfer from the lower to the upper altitudes in the mesosphere.

## 1 Introduction

The vertical propagation of atmospheric gravity waves (GWs) is known to be the main transport mechanism of momentum and energy into the upper atmosphere (Fritts and Alexander, 2003; Yiğit et al., 2016). Owing to the decrease in density with altitude, amplitudes of GWs increase exponentially if dissipation or wave breaking does not occur. GWs are excited by flows surging up mountains (e.g., Gossard and Hooke, 1975; Lindzen, 1984), fronts and jet streams (e.g., Lindzen, 1984; Fritts and Alexander, 2003; Wrasse et al., 2024), convective layers (e.g., Townsend, 1966), deep convection or thunderstorms (e.g., Taylor and Hapgood, 1988; Fritts and Alexander, 2003; Sentman et al., 2003; Yue et al., 2009; Vadas et al., 2009; Nyassor et al., 2021, 2022a, b), volcanoes (e.g., Yue et al., 2022; Figueiredo et al., 2023), typhoons (e.g., Li et al., 2022; Chou et al., 2017), earthquakes (e.g., Heale et al., 2020), solar eclipses (e.g., Paulino et al.,

2020, and references therein), and other processes that cause imbalance between the gradient of pressure and buoyancy. These waves then propagate both horizontally and vertically (Becker and Schmitz, 2003).

The vertical propagation characteristics of GWs are controlled by background temperature and wind relative to the horizontal phase speed of GWs as well as by wave dissipation due to nonlinear diffusion, viscosity, and ion drag (Yiğit et al., 2008, 2021). Depending on wave interaction with the background field, GWs can be classified as ducted, propagating, or evanescent modes (Gossard and Hooke, 1975). Some of these waves suffer from critical-level filtering, which occurs when propagating GWs encounter an equal vector of background wind, and the wave can be absorbed by the background (Heale and Snively, 2015). Otherwise, a GW can be reflected if it encounters a strong wind in the opposite direction. According to Fritts and Alexander (2003), reflected GWs from the upper- and/or lower-altitude regions can be partially ducted. Also, GWs are filtered in the middle and lower thermosphere (MLT) region during breaking. Vertically propagating GWs interact with the mean flow via the transfer of momentum and energy when breaking (Lindzen, 1981; Holton, 1982), particularly in the mesosphere. Thus, these waves significantly contribute to atmospheric circulation and dynamical fields of temperature and wind (Le Du et al., 2022; Yiğit and Medvedev, 2009).

The propagation of GWs in the horizontal and vertical directions is greatly influenced by the background wind and temperature fields (Nappo, 2013)CE5. The background fields can either hinder or enhance the vertical propagation. Doppler or thermal ducts favor the longer horizontal propagation of GWs (Bageston et al., 2011; Snively et al., 2007; Snively and Pasko, 2008), thereby hindering vertical propagation. GWs propagating vertically can either travel upward (vertical wavenumber lower than zero, $m < 0$) or downward ($m > 0$), with energy and momentum transported in either direction. A typical example is the vertical propagation of secondary GWs that result from primary GWs breaking or dissipating in the MLT (Vadas et al., 2003; Medvedev et al., 2023). During the breaking or dissipation of primary GWs, energy and momentum are released; this energy and momentum then undergo further transport upward and downward as the waves propagate (Vadas et al., 2003).

To study the vertical propagation of GWs, several observational techniques (e.g., Suzuki et al., 2013) have been employed. Observation techniques such as lidar (Suzuki et al., 2013) and radiosonde (Schöch et al., 2004; Sato and Yoshiki, 2008; Yamashita et al., 2009) have been used. In the mesosphere, Nyassor et al. (2018) used an airglow photometer to study the vertical propagation of GWs. According to Nyassor et al. (2018, and references therein), the simultaneous observation of multiple airglow emissions is one of the techniques used to investigate the vertical propagation of GWs in the mesosphere. This technique is possible if (and only if) the vertical wavelengths of the waves are larger than the thickness of the airglow emission layer (Nyassor et al., 2018, and references therein). Such observational data can be used to determine the propagation characteristics and amplitude growth of GWs (Taori et al., 2005).

Numerous studies (e.g., Fritts et al., 2006; Suzuki et al., 2013; Love and Murphy, 2016; Kaifler et al., 2020) have employed some of these observational techniques to explore the subject of GW dynamics and their momentum fluxes and potential energies. Fritts et al. (2006) investigated the momentum fluxes due to GW activities in the MLT region using wind measurements from incoherent scatter radar (ISR) at Arecibo Observatory. Using a time resolution of $\sim 50\,\mathrm{min}$, between 71 and 95 km, they quantified GW momentum flux profiles. Very high frequency (VHF) mesosphere–stratosphere–troposphere (MST) radar measurements situated near Davis Station (68.5° S, 78.0° E) were used by Love and Murphy (2016) to study the hourly averaged profiles of GW momentum fluxes between 79 and 90 km throughout the day. Love and Murphy (2016) investigated the hourly averages of momentum fluxes of the days considered (within the period from 14 December 2014 to 6 January 2015) as well as GW intermittency with altitude. Using co-located observations, Suzuki et al. (2013) investigated the vertical propagation of GWs from the lower to the upper atmosphere at the Arctic Lidar Observatory for Middle Atmosphere Research (ALOMAR; 69.31° N, 16.01° E). At the observatory, the horizontal structure of GWs is observed using a sodium (Na) airglow imager, while the ALOMAR Rayleigh–Mie–Raman (RMR) lidar and sodium lidar reveal the two-dimensional vertical structure of GWs between the stratosphere and the lower thermosphere. This coincident observation permitted the study of the horizontal and vertical characteristics of GWs and the momentum flux at the Na airglow altitude. Kaifler et al. (2020), on the other hand, used lidar with high temporal and vertical resolution to study the derived time series' absolute momentum fluxes of mountain waves at 40 km and the profile of the mean and peak absolute momentum fluxes between 10 and 80 km.

The abovementioned works have greatly contributed to quantifying the characteristics of the momentum fluxes of GWs and mountain waves (MWs) via statistical and case studies. However, none of these studies explored the aspect of how the momentum fluxes and, possibly, potential energies would behave due to the different vertical and horizontal propagation of GWs considering different background conditions. Therefore, the primary focus of this work is the behavior of the momentum flux and the potential energy of GWs under different phase or energy propagation conditions, i.e., upward, downward, or ducted.CE6 Using the vertical phase propagations of GWs, the energy propagation and, subsequently, the required parameters (e.g., vertical velocity and wavelength, potential energy, and momentum flux) can be determined.CE7

For this, an investigation was conducted on the vertical characteristics of GWs with similar ($\pm 10\,\%$ of the error mar-

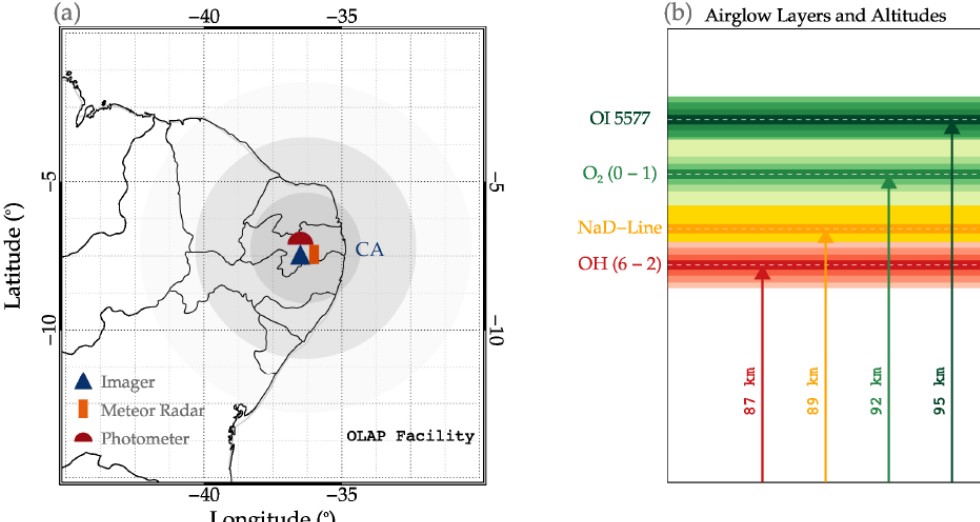

**Figure 1. (a)** The geographical location of the observatory, showing the locations of the co-located instruments, and the imager field of view at the hydroxyl (OH) airglow layer. The dark-blue triangle, orange rectangle, and red semicircle represent the positions of the all-sky imager (imager), meteor radar, and photometer, respectively (see legend). The dark-gray region indicates a radius of 256 km, the medium-gray region indicates the field of view of the imager (with a radius of 512 km), and the light-gray region shows a radius of 768 km. **(b)** Airglow emission layers and their respective altitudes. The red, gold, light-green, and dark-green colors represent the hydroxyl (OH), sodium D-line (NaD), molecular oxygen atmospheric band ($O_2(0-1)$), and green line atomic oxygen (OI 5577) emission layers, respectively. The respective dark colors with white horizontal dashed lines indicate the peak altitude of each emission layer, as indicated by the corresponding labeled vertical arrows.

gin) or nearly the same ($\pm 5\%$ of the error margin) period propagating vertically across four airglow emission layers: the atomic oxygen green line (OI 557.7 nm), molecular oxygen ($O_2$–864.5 nm) band, sodium D-line (NaD–589.0 nm), and hydroxyl (OH) (6–2) band. Following this, the phases of the wave at each layer and, consequently, the phase propagation were determined. The horizontal characteristics of the same GWs were estimated from OH all-sky images. Using the observed wind, the intrinsic parameters were also estimated. Having determined and classified these GWs as vertically propagating, the background propagation conditions and the potential energy and momentum flux at the OH and $O_2$ bands were studied. The temperature measurements employed to determine the potential energy and momentum flux were obtained using the rotational temperature at the OH and $O_2$ emission layers. The dynamics of the GWs' potential energy and momentum flux under the determined propagation condition were then studied. It was discovered that the vertical propagation of the cases selected was controlled by the background conditions imposed by the wind and temperature.

## 2 Observation and data analysis

### 2.1 Airglow

Airglow is a natural upper-atmospheric phenomenon in which light is emitted due to the de-excitation of atomic and ionic constituents from higher to lower energy levels. Physical causes of airglow include chemical reaction of neutral constituents of the upper atmosphere and reactions involving ionized constituents. Although other mechanisms do exist, these two are omnipresent and contribute to the light of the night sky (Roach, 2013). Airglow is among the atmospheric tracers used in the study of atmospheric waves. In this work, the dynamics of GWs in the mesosphere are studied using GW-modulated airglow intensities in the hydroxyl (OH (6–2), hereafter OH), sodium D-line (NaD-line, hereafter NaD), molecular oxygen ($O_2$ (0–1), hereafter $O_2$), and atomic oxygen (OI 557.7 nm, hereafter OI 5577) emission layers.

The peak altitudes of OH, NaD, $O_2$, and OI 5577 are $\sim 87$ km ($\Delta z = 8$ km), $\sim 90$ ($\Delta z = 8$ km), $\sim 92$ ($\Delta z = 8$ km), and $\sim 95$ km ($\Delta z = 8$ km), respectively. Figure 1a shows the location of the Paraíba Atmospheric Luminescence Observatory (OLAP; Observatório de Luminescência atmosférica da Paraíba in Portuguese), where the photometer, all-sky imager (hereafter, imager), and meteor radar used in this research are hosted. In Fig. 1a, the dark-blue triangle, orange rectangle, and red semicircle represent the respective positions of the imager, meteor radar, and photometer. The dark-gray region indicates a radius of 256 km, the medium-gray region indicates the field of view of the imager (with a radius of 512 km), and the light-gray region shows a radius of 768 km.

The airglow emissions, along with the peak emission (depicted using dark horizontal lines) and the altitude range ($\Delta z$) of each emission layer (depicted using respective faint

colors), are presented in Fig. 1b. The red, gold, light-green, and dark-green colors represent the hydroxyl (OH), sodium D-line (NaD), molecular oxygen atmospheric band ($O_2$), and green line atomic oxygen (OI 5577) emission layers, respectively. The respective dark colors with horizontal dashed white lines indicate the peak altitude of each emission layer, as indicated by the corresponding labeled vertical arrows. Using instruments such as airglow photometers and imagers with distinct bandpass filters, each emission layer can be observed. For this work, variations in the airglow intensity of the four emission layers, shown in Fig. 1b, are used.

### 2.1.1 Airglow photometer

The airglow photometer used for observation of the mesospheric airglow emissions is located in São João do Cariri (7.40° S, 36.31° W). The photometer is a multi-channel tilting-filter photometer (Multi-3) with five interference filters. The background continuum intensity ($R nm^{-1}$) and the line intensity in rayleigh (R) were measured to obtain the zenith sky spectrum by tilting the filters relative to their optical axes in which a scan of a wavelength of about 8 nm was made. The mesospheric component of the OI 5577 band was estimated by removing the effect of the simultaneous observation of OI 630.0 nm (hereafter, OI 6300) intensity in the ionospheric F-region component, computed as 20 % (Silverman, 1970). The temporal resolution of the observation is 2 min; thus, GWs with periods greater than 2 min can be observed. The photometer characteristics (i.e., the calibration scheme and error, the spectral resolution, and the sensitivity) can be seen in Nyassor et al. (2018) and references therein.

An observation scheme of 13 nights per month centered around the time of new moon was made and comprised more than 6 h of continuous observation time per night. The observational data used for this study extend from January 2000 to December 2007, which resulted in a total of 1051 clear-sky night observations. Details on the Multi-3 filter photometer can be found in Wrasse et al. (2004) and references therein. The database of OI 5577, $O_2$, NaD-line, and OH was analyzed to find GWs propagating with a similar or nearly the same period at each emission altitude. Among the total nights during which clear-sky night observations were possible, 389 nights present similar periods in at least two emission layers; moreover, of these 389 nights, 24 nights present similar periods in three emission layers. For this study, two GW events with the same period in all four emission layers were selected. The photometer was used for airglow intensity observation as well as observation of the rotational temperature of the $O_2$ and OH emission layers (Buriti et al., 2001).

### 2.1.2 Atmospheric band rotational temperatures derived from the OH (6–2) Meinel and $O_2$ (0–1) bands

The complex rotational band spectrum of the OH and $O_2$ emission permits the determination of mesospheric tempera-

tures by measuring the intensity distribution between various spectral lines in the bands (Innis et al., 2001). The collision frequency of OH with the neutral atmosphere near 90 km altitude has been shown to be of the order of $10^4 s^{-1}$, with a lifetime of the excited OH being around 3–10 ms (Mies, 1974). This indicates that the excited OH molecules in the rotational energy levels are in thermal equilibrium with the atmospheric ambient gas (Sivjee and Hamwey, 1987; Takahashi et al., 1998); thus, they are a good proxy for atmospheric temperature studies. The OH rotational line spectra is an open structure with a separation of 1–2 nm between the lines, which makes it easy to measure individual lines with a low-resolution (of $\sim 1$ nm) spectrometer. Further, the line intensities of most of the bands are only a function of the rotational temperature. Thus, using two lines from a single band, the rotational temperature can be estimated using the following equation (Mies, 1974):

$$T_{n,m} = \frac{E_{v'}\left(J'_m\right) - E_{v'}\left(J'_n\right)}{k_{\mathrm{B}} \ln \left[ \frac{I_n}{I_m} \frac{A\left(J'_m, v' \to J''_{m+1}, v''\right)}{A\left(J'_n, v' \to J''_{n+1}, v''\right)} \frac{2J'_m + 1}{2J'_n + 1} \right]}, \quad (1)$$

where $T_{n,m}$ is the rotational temperature estimated from two intensity lines ($I_n$ and $I_m$) from rotational levels $J'_n$ and $J'_m$ in the upper vibrational level $v'$ to $J''_{n+1}$ and $J''_{m+1}$ in the lower vibrational level $v''$; $E_v(J)$ is the energy of the level ($J, v$); $A(J'_n, v' \to J''_{n+1}, v'')$ is the Einstein coefficient for the transition from $J'_n, v'$ to $J''_n, v''$; and $k_{\mathrm{B}}$ is the Boltzmann constant.

Molecular oxygen also satisfies the local thermal equilibrium (LTE), similar to OH bands, which makes it possible to estimate the rotational temperature. $O_2$ is known to have a lifetime of more than $\sim 10$ s, making it capable of attaining the LTE. The rotational temperature can also be determined using a similar procedure to the OH rotational temperature.

### 2.2 All-sky imager

An all-sky imager was used to determine the horizontal component of the GWs observed by the photometer. Images of the OH, $O_2$, OI 5577, and OI 6300 airglow emission layers were taken by this equipment. With regard to this work, only the OH and (possibly) the $O_2$ band airglow images corresponding to the selected coincident photometer observation were used. The airglow all-sky imager is an optical instrument made of a fast ($f/4$) fish-eye lens, a telecentric lens system, a filter wheel, and a camera containing a charged-coupled device (CCD). The CCD camera has an area of 6.04 cm$^2$ with a 1024 × 1024 back-illuminated pixel array of 14 bits per pixel. In order to enhance the signal-to-noise ratio, the images were binned on the chip down to a pixel resolution of 512 × 512. The high quantum efficiency, low dark noise (0.5 electrons pixel$^{-1}$ s$^{-1}$), low readout noise (15 electron RMS, where RMS denotes the root mean square CE8 ), and high linearity (0.05 %) of this de-

vice enable it to measure airglow emissions (Medeiros et al., 2003; Nyassor et al., 2018).

## 2.3 Meteor radar

Background winds from a SKiYMET all-sky interferometric meteor radar with a two-element receiving and three-element transmitting antenna were used to observe mesospheric winds. The meteor radar operates at the same location as the photometer and makes observations between 82 and 98 km. This radar operates at a frequency of 35.24 MHz with a peak transmission power of 12 kW. The respective temporal and spatial (vertical) resolutions of this radar are typically 2 h and 3 km. The observation characteristics of the radar have been published elsewhere (Nyassor et al., 2018, and references therein, and Lima et al., 2004).

## 3 Methodology and data analysis

### 3.1 Photometer time series

The methods used to obtain the final result of the photometer data include the following: (i) preprocessing, (ii) processing, (iii) estimation of parameters, and (iv) discussion. A graphical demonstration of these procedures is shown in the flowchart in Fig. 2.

### 3.2 Preprocessing

The preprocessing stage involves four steps, as outlined in Fig. 2. Firstly, the time series is made up of the variations in wave oscillations and those due to contaminants. Hence, there is a need for visual inspection to detect any of these contaminants that appear as spikes in the time series. The contaminants can be due to artificial light sources, clouds, or astronomical lights moving across the field of view of the photometer. In Fig. 3, the hours are in universal time (UT) and span from 18:00 UT on 4 May 2004 to 04:00 UT on 5 May 2004. In Fig. 3a, a typical spike due to contamination is highlighted using the light-blue background. Furthermore, gaps are usually found in the data due to instrumental problems (although no gaps exist in these data). A criterion is set such that if the gaps or spikes in the time series cause frequent (in order of minutes) interruptions, the entire time series of the data will be disregarded. Spikes are removed from the time series CE9 . If the clean data (i.e., data without spikes) comprise a continuous observation of less than 3 h, the event on that night is disregarded. Due to the spike (the blue highlighted region) in the time series in Fig. 3a, the data are limited to the time interval from 23:00 UT on 4 May 2004 to 03:00 UT on 5 May 2004.

Clean time series with continuous observations of more than 3 h are considered for further analysis. Next, high-frequency oscillations are removed by applying a three-point running mean. Figure 3b shows the clean, smoothed (three-point mean) data. Finally, to obtain time-series data with only GW oscillations, harmonics for semidiurnal and terdiurnal tides (solid red line in Fig. 3c) are constructed using Eq. (2), as GWs are modulated by tides. CE10

$$Y = A + B_i \cos\left(\frac{2\pi(x - \phi_i)}{\tau_i}\right), \qquad (2)$$

where $A$ and $B_i$ are the unknown amplitudes, $x$ is the observation time series, $\phi_i$ is the phase, and $\tau_i$ is the period; here, $i$ represents the number of periods, in this case comprising the periods of semidiurnal ($\tau = 12$ h) and terdiurnal ($\tau = 8$ h) tides. The harmonic is subtracted from the smoothed time series to obtain a time series of the residual (purely comprised of GW oscillations). The residual (see Fig. 3d) is then used to investigate the vertical propagation of GWs.

### 3.3 Processing

In the processing stage, the first step is to inspect if the four emission layers have been modulated by the same GW(s). This is done by plotting the intensity variations. In Fig. 4a, the rotational temperature (for OH and O$_2$) and airglow intensity (for OH, Na D-line, O$_2$, and OI 5577) variations due to the GW modulations between 23:00 UT on 4 December 2004 and 03:00 UT on 5 December 2004 are presented. The temperature and intensity variations for each emission layer are defined in the respective legends. A similar variation with time was observed in the temperature, even though the temperature of the OH was higher than that of O$_2$. In the case of airglow intensities, well-defined similarities due to the presence of GWs were observed. The variations in the temperature and intensity presented a down-phase progression with time except for OH intensity variations.

The temperature and intensity variations due to GW modulation for a second case observed between 18:30 and 23:30 UT on 1 May 2005 are presented in Fig. 4b. Similar oscillations can be seen in the variations in the temperatures and the intensities. Unlike the case presented in Fig. 4a, the O$_2$ and OH temperature variations present an almost upward phase progression. Similar phase characteristics can be found in the variation in the NaD and OH airglow intensities. In contrast, the intensity variations in OI5577 and O$_2$ present a downward phase progression.

In the processing stage, the residuals were subjected to Lomb–Scargle periodogram and wavelet analysis to determine the dominant periods in the time series of each emission layer. In Figs. 5 and 6, the clean and smoothed time series, the residual, and the Lomb–Scargle periodogram for all intensities and rotational temperatures of the emission layers for the respective GW events on 4–5 December 2004 and 1 May 2005 are presented. In Figs. 5a and 6a, the intensity and temperature time series (similar to Fig. 3c) are shown, whereas the residuals are presented in Figs. 5b and 6b. The solid red line in Figs. 5a and 6a indicates the tidal harmon-

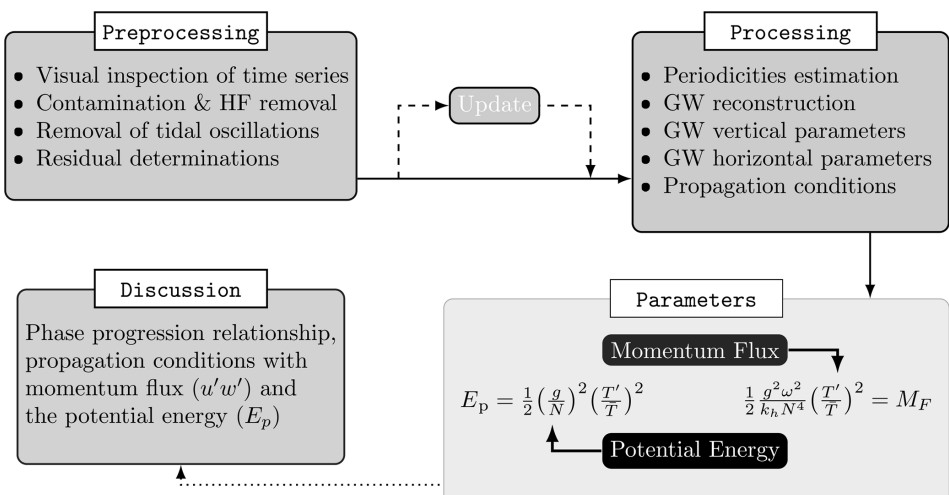

**Figure 2.** Flowchart showing the airglow photometer data processing procedures and GW characterization. The procedure includes preprocessing, processing, parameterization, and discussion.

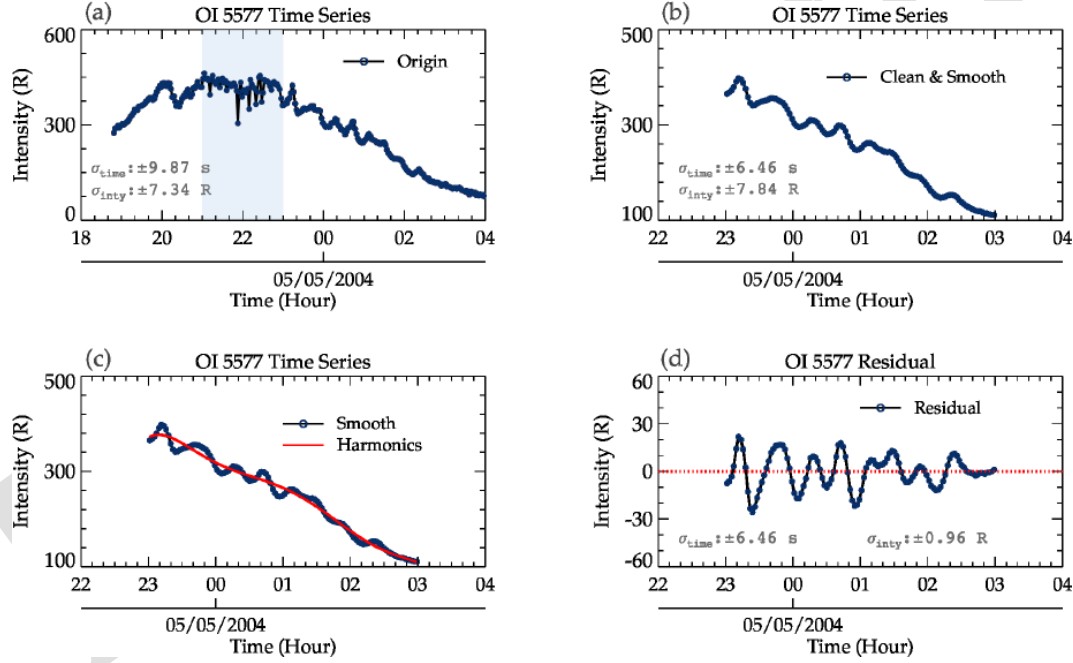

**Figure 3.** A step-by-step procedure of the preprocessing stage for photometer data. Panel **(a)** is the original OI 557.7 nm time series, with the light-blue shading indicating the parts of the time series with spikes. In panel **(b)**, the clean and smoothed time series is presented. The harmonics of tides, semidiurnal and terdiurnal (solid red line), are constructed and shown in panel **(c)**. The residual (difference between the harmonics and the clean and smoothed data) in panel **(c)** is shown in panel **(d)**.

ics. Subpanels (iii) and (vi), with the gray background, represent the respective rotational temperatures of the $O_2$ and OH emission as well as their residual and Lomb–Scargle periodogram. Subpanels (i), (ii), (vi), and (v) represent the clean and smoothed, residual, and Lomb–Scargle periodogram of the OI 5577, $O_2$, NaD, and OH intensities.

In Fig. 5c, the Lomb–Scargle periodogram of each intensity and the respective $O_2$ and OH rotational temperatures are presented. As mentioned earlier, at least two similar or nearly the same periods present in each emission layer are selected. The chosen periods are demarcated by vertical dashed lines. It can be seen that three similar periods were present; however, only two were used in the follow-up analysis. This is because the differences in the emission layers are quite large, especially for the OI 5577 emission intensity and the OH rotational temperature.

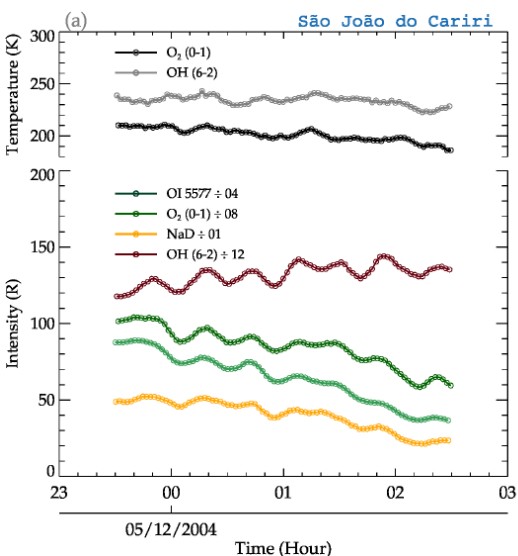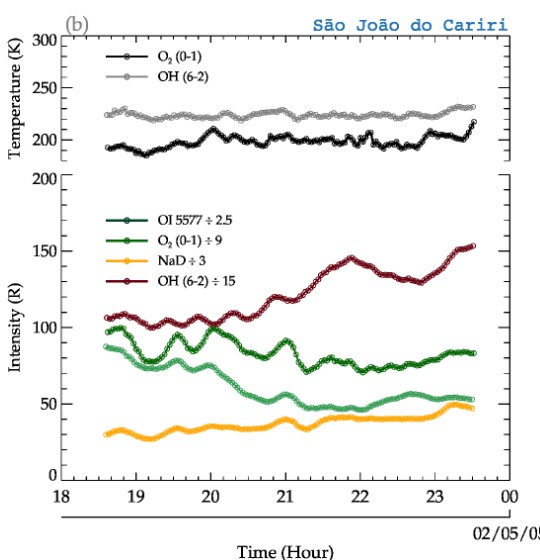

**Figure 4.** Rotational temperature and airglow intensity variations within the periods from **(a)** 23:00 UT on 4 December 2004 to 03:00 UT on 5 December 2004 and **(b)** 18:30 to 23:30 UT on 1 May 2005.

Similar to Fig. 5, Fig. 6 presents the processing stage of the GW event on 1 May 2005. For this event, three dominant periods were detected. For the first period, OI, $O_2$, and NaD present periods of 0.535477 h (32.13 min), whereas OH presents a period of $\sim 0.503125$ h (30.19 h) CE11 . The second and third dominant periods of $\sim 0.720896$ h (43.25 min) and $\sim 1.10274$ h (66.16 min), respectively, are present in all of the emission layers. The periods are indicated by the vertical black dashed lines.

From the above description, at least two dominant peaks are chosen, used to reconstruct new harmonics, and plotted over the residual. The new harmonics are then normalized and plotted in order of increasing altitude, i.e., 87, 89, 92, and 95 km. Note that the rotational temperatures of OH (6–2) and $O_2$ were also subjected to Lomb–Scargle and wavelet analysis in order to verify that the temperature was also modulated by the observed GWs with respect to the intensity. From the reconstructed time series and according to their altitude, the phases ($\phi$) of the GWs at each altitude were determined.

The phase ($\phi$) is estimated from Eq. (2) and is given in decimal hours. However, the phase needs to be estimated considering the start time of the data being used. This is done in order to give the phase in relation to the start time of the time series. This is carried out by adding the individual dominant period in the time series until it corresponds to the first hour that concurs with the start time of the data of each emission layer. For instance, the phase of the 4 December 2004 OI 5577 GW event is 0.0744475 h for GWs with a period of 0.424451 h ($\sim 25.47$ min). To establish the phase in a form related to the time of observation, the period was added to the phase until 23.419253 h (23:25:00 UT) was attained. This is the time corresponding to the first hour of the time series used. A similar procedure was applied to the intensities of the other emission layers. In Table 1, the result of the phases of the two waves selected for Event no. 01 and the three waves selected for Event no. 02 are presented. The phase shifts were established from the phases determined from each individual emission layer. The phase differences were then estimated between each two consecutive layers as well as the first and the fourth layers.

The error associated with each emission layer has been established to evaluate the impact on the result obtained. The error was assessed by estimating the standard error in the original data, the smoothed data, and the harmonics. It is important to mention that the estimated standard error of the mean ($\sigma_M$) for OI 5577, $O_2$, NaD, and OH intensities ($\sigma_{M_I}$); temperature ($\sigma_{M_T}$); and time ($\sigma_{M_{time}}$) are presented in Table 2. The standard errors are estimated for original data, clean and smoothed data, the residual, and the harmonics.

From Table 2, the estimated errors in the time, intensities, and temperature of the original data; the clean and smoothed data; the residual; and the harmonics are presented. In general, the errors associated with the original data are higher than those of the smoothed data and the residual. These values are, however, less than the measurement errors of the intensities, which are of the order of 5 %, whereas for rotational temperature, errors are 2–3 K for $O_2$ and 4–5 K for OH (Wrasse et al., 2004). The error associated with the fit was evaluated by estimating the cross-correlation between the time series of the residual of the intensities and temperature and their respective harmonics, and these time series are indicated in Figs. 7 and 8. The cross-correlations of the intensities (i.e., their residuals and harmonics) are 0.76, 0.79, 0.65, and 0.77 for the OI, $O_2$, NaD, and OH intensities, respectively. For the temperature residuals and harmonics, values for $O_2$ and OH are 0.52 and 0.67, respectively.

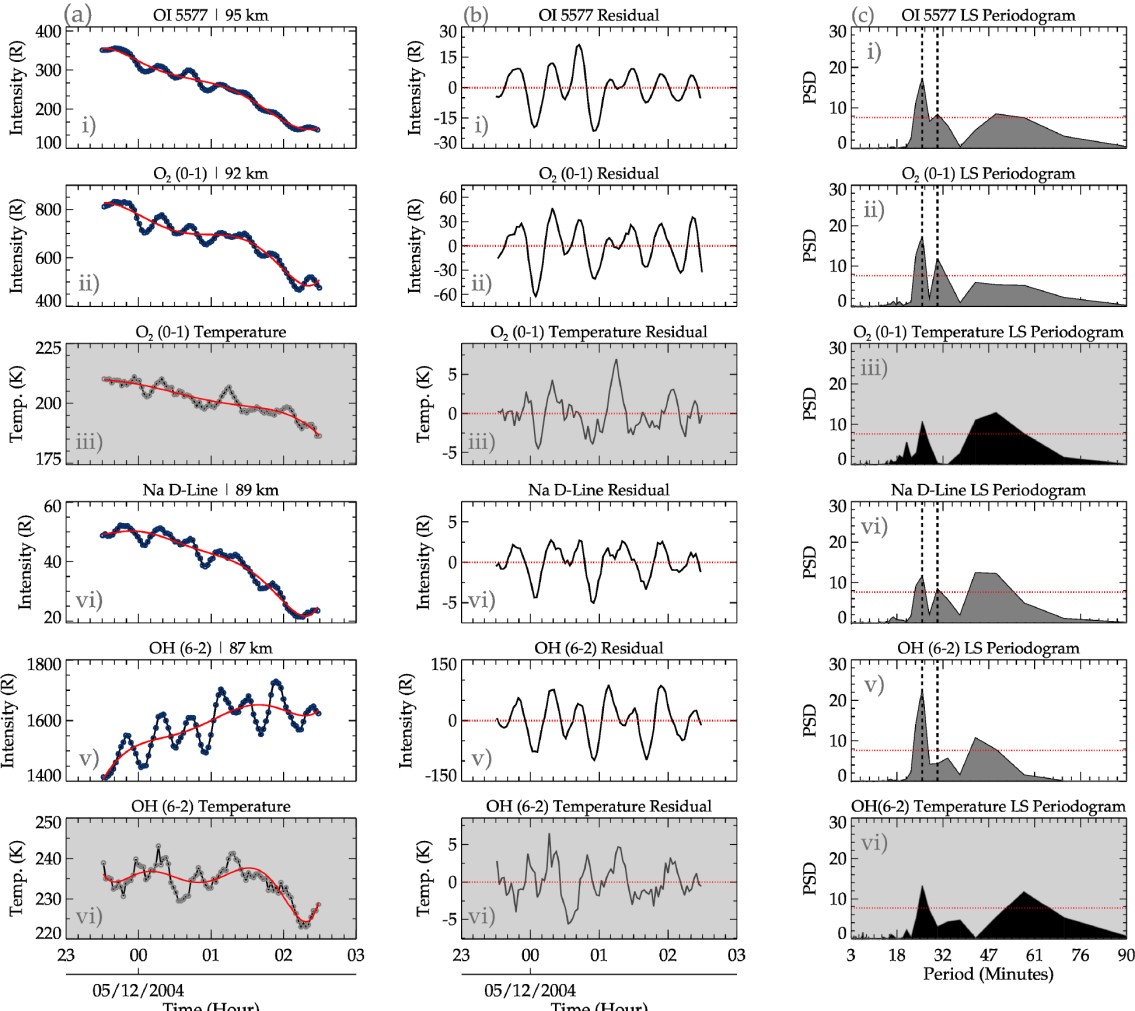

**Figure 5.** A detailed description of the processing stage of the preprocessed data (obtained from Fig. 3). The reconstructed harmonics of the tidal wave oscillations (solid red line) using the dominant periods of 12 and 8 h, the intensity variations due to GW modulation for each airglow emission layer, and the GW modulated rotational temperature for the $O_2$ and OH airglow emission layer are presented in panel **(a)**. The intensity residuals of only GW oscillations are shown in panel **(b)**. In panel **(c)**, the Lomb–Scargle periodogram result of each emission layer intensity and the rotational temperatures of $O_2$ and OH are presented. The horizontal dotted red lines in panel **(c)** represent the 95 % significance level.

Using the differences in phase and altitude between each of the two consecutive emission layers, the average vertical wavelength ($\lambda_z$) of the wave is given by Nyassor et al. (2018):

$$\lambda_z = \frac{V_z}{\tau}. \tag{3}$$

Here, $V_z = \Delta d / \Delta \phi$ is the vertical velocity, with $\Delta d$ being the difference between the higher and lower emission layers and their respective phases (denoted by $\Delta \phi$), and $\tau$ is the period. A typical result obtained from the procedures in the processing stage is presented in Figs. 7 and 8.

### 3.4   Parameters

In the parameter stage, the potential energy ($E_p$) and momentum flux ($u'w'$) of the GWs were estimated. The potential energy is estimated using the approach of Narayanan et al. (2024):

$$E_p = \frac{1}{2}\left(\frac{g}{N}\right)^2 \left(\frac{T'}{\overline{T}}\right)^2, \tag{4}$$

where $g$ is the gravitational acceleration, $N$ is the Brunt–Väisälä frequency, $\overline{T}$ is the background temperature, and $T'$ is the temperature variation due to GW perturbation. The Brunt–Väisälä frequency is defined according to Wrasse

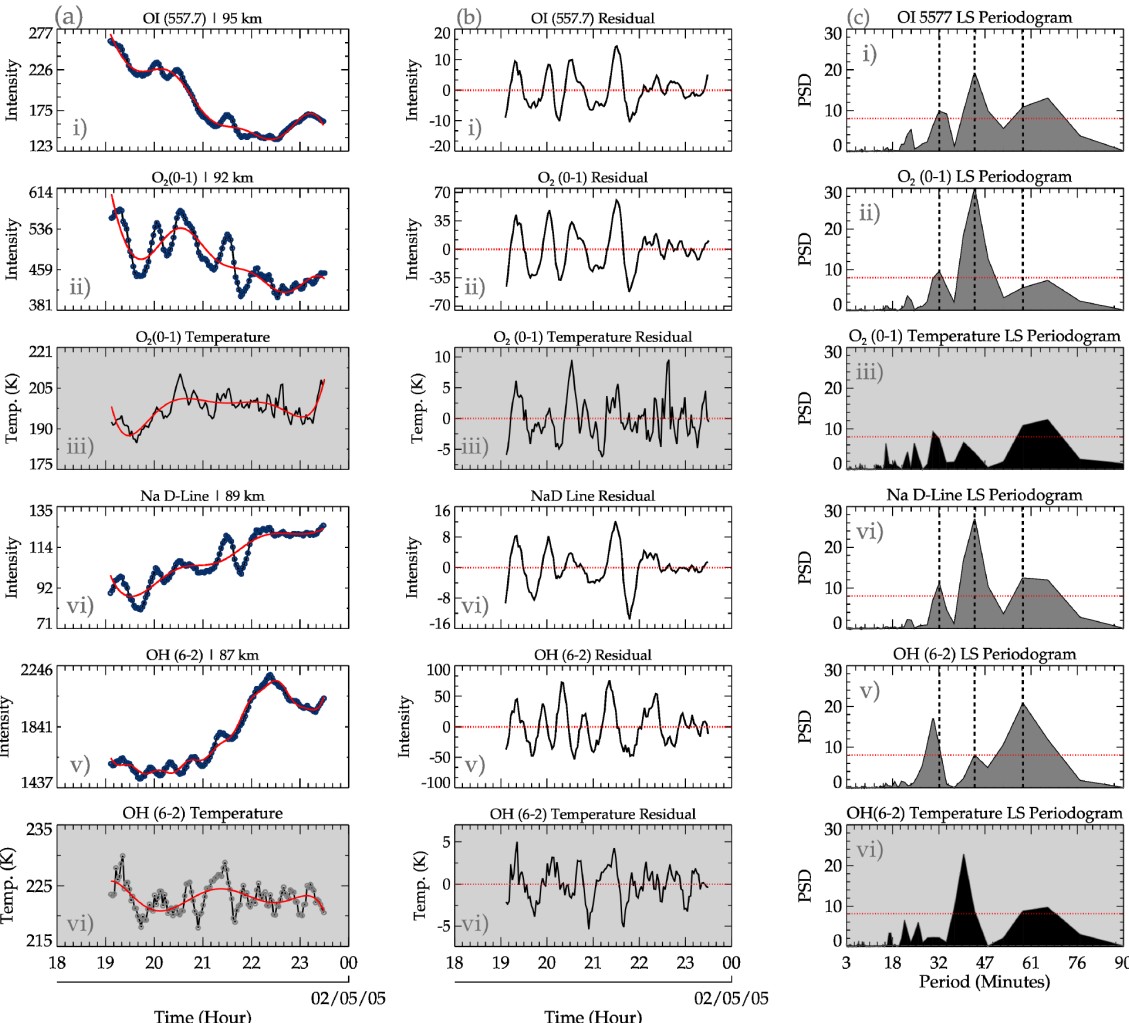

**Figure 6.** Similar to Fig. 5 but for the event between 18:30 and 23:30 UT on 1 May 2005.

**Table 1.** Estimated phases (in hours) in each emission layer for the 4–5 December 2004 and 1 May 2005 GW events. The subscripts $\tau$ (of $\phi$) indicate the phase of the corresponding periods.

| Events | Phases $\phi$ | OI | $O_2$ | NaD | OH |
|---|---|---|---|---|---|
| Event no. 01 | $\phi_{\tau=25.24}$ (h) | 23.419253 | 23.417403 | 23.829630 | 23.419026 |
| | $\phi_{\tau=38.00}$ (h) | 23.259999 | 23.312784 | 23.293561 | 23.683039 |
| Event no. 02 | $\phi_{\tau=31.64}$ (h) | 18.880139 | 18.888510 | 18.856413 | 18.844777 |
| | $\phi_{\tau=43.25}$ (h) | 18.560785 | 18.573995 | 18.565454 | 18.565454 |
| | $\phi_{\tau=58.43}$ (h) | 18.570727 | 18.548215 | 18.465382 | 18.379787 |

et al. (2024) as follows:

$$N = \left( \frac{g}{\theta} \frac{d\theta}{dz} \right)^{1/2}, \qquad (5)$$

where

$$\theta = T (P/P_0)^{R/c_p}. \qquad (6)$$

Here, $\theta$ is the potential temperature; $p$ and $p_o$ are the pressure and reference pressure, respectively; $R$ is the gas constant; and $c_p$ is the heat capacity at constant pressure. During this wave event, the SABER (Sounding of the Atmosphere using Broadband Emission Radiometry) instrument, aboard the TIMED (Thermosphere Ionosphere Mesosphere Energetics Dynamics) satellite, made an overpass $\sim 735$ km from the OLAP observation site. The temperature profiles

**Table 2.** Associated mean standard errors in the time series of the observation hour ($\sigma_{M_{\text{time}}}$) in seconds, intensity ($\sigma_{M_{\text{I}}}$) in rayleigh, and temperature ($\sigma_{M_{\text{T}}}$) in kelvin for the events on 4 December 2004 and 1 May 2005.

| Errors | Event no. 01 | | | | Event no. 02 | | | |
|---|---|---|---|---|---|---|---|---|
| | OI | $O_2$ | NaD | OH | OI | $O_2$ | NaD | OH |
| *Original data* | | | | | | | | |
| $\sigma_{M_{\text{time}}}$ (s) | ± 09.871 | ± 09.871 | ± 09.871 | ± 09.871 | ± 06.786 | ± 06.786 | ± 06.786 | ± 06.786 |
| $\sigma_{M_{\text{I}}}$ (R) | ± 07.339 | ± 14.745 | ± 04.076 | ± 23.243 | ± 03.350 | ± 04.316 | ± 01.110 | ± 18.508 |
| $\sigma_{M_{\text{T}}}$ (K) | | ± 01.728 | | ± 03.132 | | ± 00.471 | | ± 00.186 |
| *Usable data range* | | | | | | | | |
| $\sigma_{M_{\text{time}}}$ (s) | ± 05.604 | ± 05.604 | ± 05.605 | ± 05.604 | ± 06.786 | ± 06.786 | ± 06.786 | ± 06.786 |
| $\sigma_{M_{\text{I}}}$ (R) | ± 06.958 | ± 10.911 | ± 01.052 | ± 07.998 | ± 03.350 | ± 04.316 | | ± 01.110 TS1 |
| $\sigma_{M_{\text{T}}}$ (K) | | ± 00.670 | | ± 00.413 | | ± 00.471 | | ± 00.186 |
| *Residual* | | | | | | | | |
| $\sigma_{M_{\text{time}}}$ (s) | ± 05.604 | ± 05.604 | ± 05.605 | ± 05.604 | ± 06.786 | ± 06.786 | ± 06.786 | ± 06.786 |
| $\sigma_{M_{\text{I}}}$ (R) | ± 00.964 | ± 02.537 | ± 00.203 | ± 05.024 | ± 00.474 | ± 02.114 | ± 00.410 | ± 02.727 |
| $\sigma_{M_{\text{T}}}$ (K) | | ± 00.236 | | ± 00.253 | | ± 00.289 | | ± 00.172 |
| *Harmonics* | | | | | | | | |
| $\sigma_{M_{\text{H}}}$ (R) | ± 00.732 | ± 01.993 | ± 00.132 | ± 03.850 | ± 00.411 | ± 01.901 | ± 00.380 | ± 02.256 |

obtained from the SABER sounding are used in the study of the propagation conditions of each selected GW event in this work. The measured pressure from the SABER observation was used in the determination of the potential temperature; here, $K/c_p = 0.286$. Using a first-order derivative procedure in interactive data language (IDL) (Bowman, 2006), the $d\theta/dz$ profile was determined. The potential temperatures at the peak altitudes of OH and $O_2$, which are $\sim 87$ and $\sim 92$ km, were chosen. However, the IDL procedure requires three data points in order to be able to compute the $d\theta/dz$. As the temperatures of the OH and $O_2$ layers are known, the temperature of OI 5577 was inferred from the SABER observation to calculate $\theta$ and then used to create a time series (which was kept constant) to attain the required input. Thus, a constant time series of $\theta$ was created for the OI 5577 emission layer.

The zonal and meridional momentum fluxes of the GWs are determined, by adapting the approach of Suzuki et al. (2007) and Vargas et al. (2009), as follows:

$$M_{\text{Fzon}} = \rho_0 \langle u'w' \rangle = -\rho_0 \frac{1}{2} \frac{km\omega^2}{k_{\text{H}}^2} \frac{g^2}{N^4} \left( \frac{T'}{\overline{T}} \right)^2,$$

$$M_{\text{Fmer}} = \rho_0 \langle v'w' \rangle = -\rho_0 \frac{1}{2} \frac{lm\omega^2}{k_{\text{H}}^2} \frac{g^2}{N^4} \left( \frac{T'}{\overline{T}} \right)^2. \quad (7)$$

Here, $\rho_0$ represents the density at the emission layers; $k_{\text{H}}^2 = k^2 + l^2$ is the horizontal wavenumber, with $k$ and $l$ being the respective zonal and meridional wavenumbers; $m$ is the vertical wavenumber; $\omega$ is the intrinsic frequency; $g$ is the gravitational acceleration; and $N$ is the Brunt–Väisälä fre-

quency. The density, $\rho_0$, used in Eq. (7) was obtained from a SABER sounding close to the observation site during each GW event. The intrinsic frequency $\omega$ can be estimated from the following expression: $\omega_0 - kU - lV$, where $\omega_0 = 2\pi/\tau$. $U$ and $V$ are the respective zonal and meridional wind speed in the direction of the wave at each peak emission altitude. The horizontal wavenumber, $k_{\text{H}}$, was estimated from the horizontal wavelength, $\lambda_{\text{H}}$, estimated by Eq. (A5) in Appendix A1, using the relation $k_{\text{H}} = 2\pi/\lambda_{\text{H}}$. The term $T'/\overline{T}$ is the relative temperature perturbation, $T'$ is the GW-induced temperature variation, and $\overline{T}$ is the background temperature. The total momentum flux ($M_{\text{F}}$) of the GW is given by Vargas et al. (2007):

$$M_{\text{F}} = \rho_0 \langle u'w' \rangle = -\rho_0 \frac{1}{2} \frac{g^2}{N^4} \frac{m}{k_{\text{H}}} \omega^2 \left( \frac{T'}{\overline{T}} \right)^2. \quad (8)$$

Estimating the potential energy ($E_{\text{p}}$) and the momentum flux ($M_{\text{F}}$) of GWs depends on observed temperature and wind data. As mentioned earlier, the rotational temperatures from photometer observations were used for $M_{\text{F}}$ and $E_{\text{p}}$ estimation.

### 3.5 All-sky image preprocessing and spectral analysis

To determine the horizontal parameters of the selected events, images from the co-located all-sky imager at São João do Cariri were used. The methodology used for all-sky image processing and GW parameter estimation calls for the validated and updated image preprocessing and spectral analysis (iPreSA) routine of Wrasse et al. (2024), which is capable of

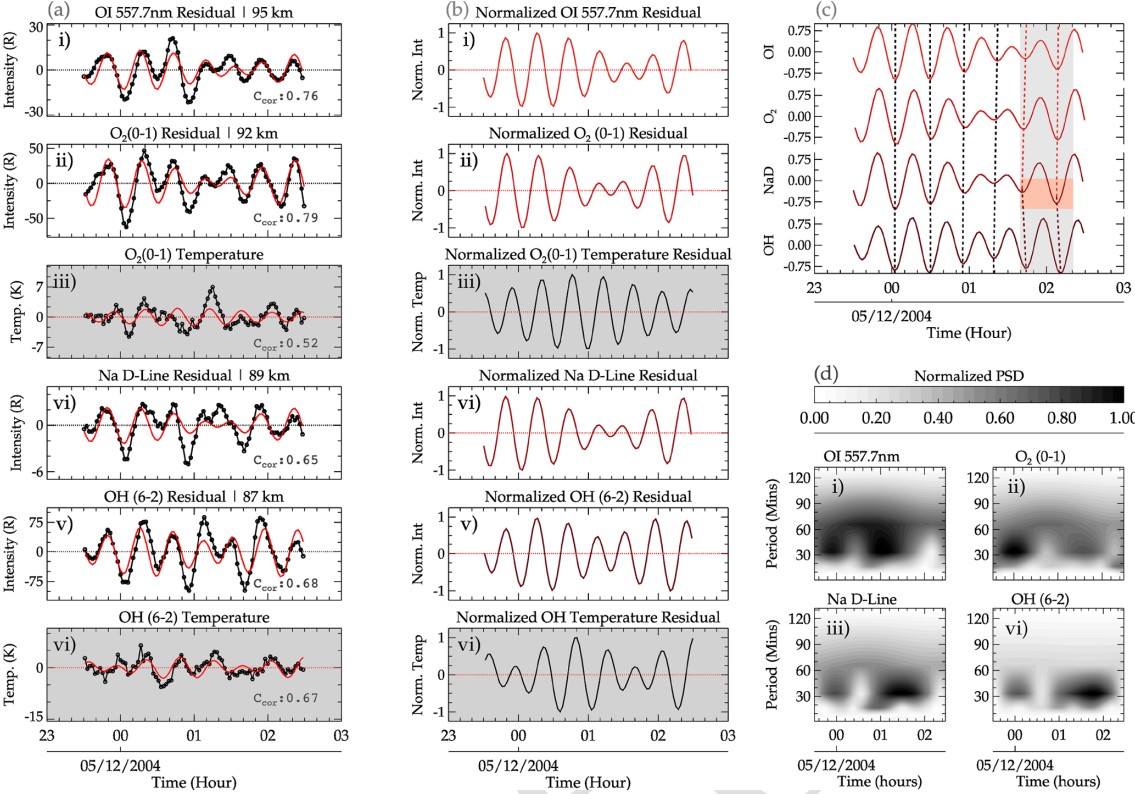

**Figure 7.** A detailed description of the processing stage of the preprocessed data (obtained from Fig. 3). The reconstructed harmonics of the GW oscillations (solid red line), using the dominant periods determined by wavelet and Lomb–Scargle periodogram analysis, and the residuals for each airglow emission layer and the rotational temperature are presented in panel **(a)**. The corresponding normalized residuals are shown in panel **(b)**. Using the normalized residuals in panel **(c)**, the phase propagation of the GW oscillation at each emission layer altitude is determined using the vertical slanted dotted lines. The gray background emphasizes the vertical red dashed lines, whereas the turning point of the phase lines of the dashed lines are highlighted by the light-red background. In panel **(d)**, the wavelet analysis result of each emission layer is shown.

preprocessing original airglow images and retrieving GWs' horizontal characteristics. The preprocessing aspect includes the following: (a) calibration of the original images (i.e., to align the image with the geographic coordinates), (b) re-
5 moval of stars to reduce their effect on the wave spectrum at high frequencies (Maekawa, 2000), and (c) correction of the curvature effect of the CCD camera by mapping the original images onto new coordinates that relate distances between pixels in the image to physical distances in the airglow
10 layer with the zenith at the origin. Other minor corrections implemented in the preprocessing stage involve the estimation of the intensity fluctuation fraction and the application of a high-pass filter and a weighting function. The current version of the preprocessing also incorporates correction of
15 the van Rhijn effect and atmospheric extinction as well as the removal of the Milky Way (Kubota et al., 2001; Wrasse et al., 2024).

  The wave characteristics in the preprocessed airglow images are obtained using a two-dimensional spectral analysis
20 technique. The underlying concept of this technique is the two-dimensional discrete Fourier transform (2D-DFT), rep-

resented mathematically as follows:

$$\mathcal{F}(k,l) = \sum_{x=0}^{m-1}\sum_{y=0}^{n-1}\left(c^{-i\frac{2\pi xk}{m}}\right)\left(c^{-i\frac{2\pi yl}{n}}\right)f(x,y). \qquad (9)$$

Here, $\mathcal{F}(k,l)$ is the Fourier transform of the function, where $k$ and $l$ are the respective zonal and meridional wavenum- 25 bers, and $m \times n$ is the dimension of the analyzed image. The cross-spectrum between two successive images is estimated from the amplitude and the phase of the waves and, subsequently, the horizontal wavelength ($\lambda_H$), the period ($\tau_H$), the phase speed ($c_H$), and the propagation direction ($\phi_H$). The 30 subscript "H" indicates horizontal. A step-by-step description of the spectral analysis of Wrasse et al. (2007), which was used to estimate the horizontal parameters of the observed GWs (Figueiredo et al., 2018; Wrasse et al., 2024), is presented in Appendix A1. From the horizontal wave- 35 length ($\lambda_H$), the zonal ($k$) and meridional ($l$) wavenumbers were determined and used in Eqs. (7) and (8) to estimate the momentum flux. An important condition considered in the selection of the horizontally propagating GWs is that the

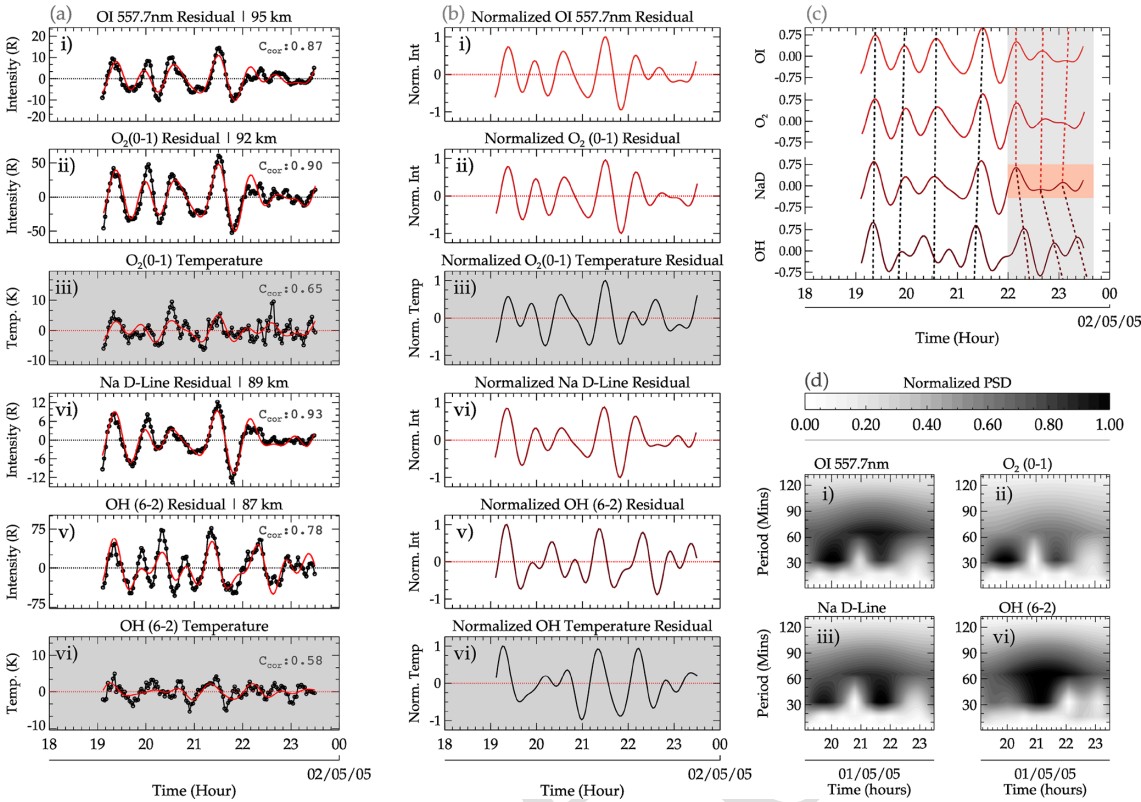

**Figure 8.** Same as Fig. 7 but for the GW event on 1 May 2005.

period must be equal or similar to the period of the vertical component observed in the photometer data. A summary of the keogram spectral analysis and the obtained results is given in Figs. A1–A5 in Appendix A1.

## 4    Results

The results of the two selected cases outlined in Sect. 3 are presented in Figs. 7 and 8. Event no. 01 and Event no. 02, which occurred in São João do Cariri on 4 December 2004 and 1 May 2005, respectively, are presented. For selection, these cases must have satisfied the following criteria:

1. GWs with similar or the same periods must be observed in all four emission layers.

2. Similar or the same periods observed in the four airglow intensity variations must be present in the OH and $O_2$ rotational temperatures.

3. Similar or the same periods observed in criterion no. 1 must be present in the OH and possibly $O_2$ all-sky images.

4. The periods of GWs in each emission layer must be present in the time series for 3 h or more.

Besides selecting these two GW events due to the presence of similar or the same periods in the four emission layers, these events permit the exploration of the dynamics of GWs using (i) observed variables in the MLT region at high temporal resolution and (ii) derived momentum flux. To achieve this, all four criteria must be satisfied. Most of the observed cases that met criterion no. 4 do not satisfy either criterion no. 1 or no. 2. This is to say that most of the cases have similar or the same period in three emission layers and one rotational temperature similar to the work of Nyassor et al. (2018). On the other hand, the majority of the events have similar or the same period in just two emission layers.

### 4.1   GW event on 4–5 December 2004

In Fig. 7a (the 4 December 2004 GW event), the reconstructed GW oscillations (solid red line) using the observed periods determined in the OI, $O_2$, NaD, and OH emission layers plotted over the residual are presented. The dominant periods used in the reconstruction of the waves in GW Event no. 01 are 00.42 h (25.47 min) and 00.50 h (30.29 min). The wave of $\tau = 25.47 \pm 02.40$ min was observed in all four emission layers. For the second wave of this event, two similar periods, $\tau = 30.29$ min and $\tau = 33.47$ min, were observed. The wave with $\tau = 30.29$ min was detected in the OI, $O_2$, and NaD emission layers. However, the observed period in

the OH layer was 0.55 h (33.47 min). Even though this period may differ from the period determined in the other emission layers, its deviation was not outside the error margin of the estimation and, thus, was considered. The period $\tau = 33.47 \pm 03.16$ min is used in subsequent analysis. The error ranges were within $\pm 10\%$ of the estimated periods. The consideration was based on the fact that these are observed periods that were determined under background wind conditions.

In Fig. 7b, the normalized residual of the reconstructed time series presented for each emission intensity and rotational temperature is shown. The normalization is computed so as to standardize the range of variations in all of the emission layers. From the normalized intensity variations, Fig. 7c was produced, from which the phase differences were determined. From the phase progression, an upward phase propagation, indicated by the first four vertical dashed black lines, was observed. For these four vertical dashed lines, a sharp upward phase propagation (nearly vertical) was observed across the four emission layers of OI, $O_2$, NaD, and OH between 23:00 UT on 4 December 2004 and 01:30 UT on 5 December 2004. Between the hours of 01:30 and 02:20 UT on 5 December 2005, a steep upward phase propagation can be seen between the OI, $O_2$, and NaD emission layers, after which a downward phase propagation was then observed between NaD and OH. The turning point of the phase lines of the dashed red lines is highlighted by the light-red background, whereas the vertical dashed red lines are emphasized by the gray background.

The observed periods estimated using wavelet analysis are presented in Fig. 7d. The strong presence of a range of wave periods between 25 and 90 min was observed in all four emission layers with dominant periods of $25.47 \pm 02.40$ min (all emission layers), 30.29 min (for OI, $O_2$, and NaD), and 33.47 min (OH). As mentioned earlier, only the period $33.47 \pm 03.16$ min is considered in the subsequent analysis. These dominant periods were also present in the Lomb–Scargle periodogram, as shown in Fig. 5. The plots Fig. 5d are normalized to standardize the variations in the individual emission layers with the scale defined in the color bar.

A peak selection procedure considering the power spectral densities (PSDs) was used to detect the significant periods. It is important to emphasize that GW events with the same periods or similar periods (within $\pm 5\%$ deviation) were chosen. The deviations were included in the choice of periods, as they are observed periods that are susceptible to the background wind. Using these periods and Eq. (2), the time series (solid black line with open circles) for each emission layer were reconstructed (solid red line), as shown in Fig. 7a. Unlike Fig. 3, the residuals of all of the emission layers are plotted, including the rotational temperatures of OH and $O_2$. Similar to Figs. 5 and 6, the rotational temperatures are presented in the plots with a gray background in Fig. 7a (subpanels iii and vi). In subpanels (i)–(vi) of Fig. 7b, the normalized re-

constructed residual for each emission layer (including the rotational temperatures of OH and $O_2$) is presented.

From the reconstructed time series, a similarity can be seen in all of the emission layers, which serves as an indication of similar GWs propagating through these layers. Even though similarities exist in the periodicities, some degree of variations can be seen. These variations can be attributed to the variations in the background wind, as the result obtained in the Lomb–Scargle periodogram and the wavelet analysis are observed periods. It is worth mentioning that the time series of the rotational temperature has also been subjected to all of the abovementioned procedures to confirm that similar waves observed in the intensity are also present in the temperature. In Fig. 7c, the normalized reconstructed time series intensity is plotted in ascending order of altitude, from which the phase progression of the waves with altitude is determined. Using the vertical dashed lines, the phase progressions are determined.

## 4.2 GW event on 1 May 2005

In Fig. 8 (which is similar to Fig. 7), panels (a)–(d) have the same arrangement. However, this event started at 18:00 UT and finished at 23:30 UT on 1 May 2005. In all of the subpanels of Fig. 8a, the reconstructed time series (solid red lines) using Eq. (2) are plotted with the corresponding residuals of the airglow intensity variations as well as the $O_2$ and OH rotational temperatures. To assess how best the reconstructed time series of each emission intensity variations (and the rotational temperatures of OH and $O_2$) fits the corresponding residual, the cross-correlation ($C_{cor}$) was estimated and is presented in each subpanel of Fig. 8a. In Fig. 8b, the respective normalized intensities and rotational temperatures are presented.

The periods determined in each emission layer using wavelet analysis are presented in Fig. 8d. In subpanels (i), (ii), (iii), and (iv) of Fig. 8d, the respective spectrograms indicating the PSDs relating the intensity of the periodicities of the wave to the time of occurrence for the OI, $O_2$, NaD, and OH emission layers are presented. The PSD for all of the airglow emission layers has been normalized. The scale of the variations is defined by the color bar. A broad spectrum of high PSD was observed for the periods, extending from 30 to $\sim 90$ min throughout the entire observation window with a peak centered around the early hours of the observation and through the middle and the later time of the observation (especially for the OH emission). A summary of the wave parameters of the photometer observations is presented in Table 3.

In Fig. 8c, the phase lines indicating the phase propagation across the four emission layers are presented. Between the hours of 19:30 and $\sim 22:00$ UT, a steep upward phase propagation extends from the OH, through NaD and $O_2$, and to the OI emission layers. After 22:00 UT until 23:30 UT, the phase propagation was upward between the NaD and OI emission

**Table 3.** Summary of the selected GW events.

| Events | Photometer | | | All-sky imager | | Parameters | |
|---|---|---|---|---|---|---|---|
| | $\tau_z$ (min) | $V_z$ (m s$^{-1}$) | $\lambda_z$ km | $\tau_H$ (min) | $\lambda_H$ (km) | $M_F$ (m$^2$ s$^{-2}$) | $E_p$ (J kg$^{-1}$) |
| | | | Event no. 01 | | | | |
| $\tau_1(O_2)$ | $25.47 \pm 02.40$ | inf | inf | $23.10 \pm 1.20$ | $91.00 \pm 6.30$ | $03.50 \times 10^{-5}$ | 44.33 |
| $\tau_1(OH)$ | | | | | | $01.24 \times 10^{-5}$ | 11.53 |
| $\tau_2(O_2)$ | $33.47 \pm 03.16$ | $05.28 \pm 01.25$ | $10.60 \pm 02.50$ | $33.60 \pm 1.70$ | $135.41 \pm 11.59$ | $419.50 \times 10^{-3}$ | 44.33 |
| $\tau_2(OH)$ | | | | | | $07.30 \times 10^{-3}$ | 11.53 |
| | | | Event no. 02 | | | | |
| $\tau_1(O_2)$ | $31.64 \pm 02.93$ | $62.84 \pm 11.43$ | $119.31 \pm 21.70$ | $28.90 \pm 1.40$ | $255.90 \pm 17.60$ | $13.80 \times 10^{-3}$ | 104.62 |
| $\tau_1(OH)$ | | | | | | $00.30 \times 10^{-3}$ | 7.13 |
| $\tau_2(O_2)$ | $43.25 \pm 04.06$ | $108.08 \pm 19.55$ | $280.50 \pm 25.38$ | $46.20 \pm 2.30$ | $237.00 \pm 14.40$ | $34.80 \times 10^{-3}$ | 104.62 |
| $\tau_2(OH)$ | | | | | | $00.90 \times 10^{-3}$ | 7.13 |
| $\tau_3(O_2)$ | $58.43 \pm 05.49$ | $11.64 \pm 02.17$ | $40.80 \pm 07.62$ | $61.03 \pm 3.10$ | $454.40 \pm 26.20$ | $95.40 \times 10^{-3}$ | 104.62 |
| $\tau_3(OH)$ | | | | | | $01.50 \times 10^{-3}$ | 7.13 |

layers, whereas phase propagation was downward between NaD and OH.

## 5 Discussion

### 5.1 Phase propagation

As presented in Sect. 4, two events of similar periods were selected. For Event no. 01, two dominant periods were detected; however, the estimated phase propagation associated with the first period indicates little or no phase change, implying near-vertical propagation, which is indicative of a possible ducted wave. For Event no. 02, three dominant observed periods were determined. In Fig. 9, the phase leads and lags between the four emission layers are presented. This figure is intended to determine how much these GWs with similar periods propagating through the emission layers lead or lag the preceding or succeeding layers using the phase shifts.

#### 5.1.1 Event no. 01

Based on the phases of the GWs of Event no. 01, OH leads NaD by 8.60 min, NaD leads $O_2$ by 1.21 min, and $O_2$ lags OI by 3.25 min. A consistent phase lead can be observed from OH through NaD to $O_2$ except between $O_2$ and OI, where a phase lag was observed. The phase lag observed between the $O_2$ and OI emission layers was induced by the background wind (due to a shear). Based on the average wind between 23:30 and 02:30 UT in the direction of the wave, a wind shear existed between 80 and 98 km, as shown in subpanel (iii) in Fig. 10a. This contributed to the phase shift in the waves. As seen in Fig. A6, there was a change in the zonal wind direction from east to west above the OH emission layer within the observation hour of the event. Above the NaD emission layer, the zonal wind became predominantly westward and peaked in the $O_2$ layer. Similarly, the meridional wind also exhibited a change in direction from north to south within the NaD layer during the observation window of the GW event. Despite this phase lag, the mean phase propagation of these GWs shows that OH leads OI by $\sim 06.58$ min. Using this phase information and the period, Fig. 9 was produced. These figures are used to evaluate the phases of the GWs in these two (i.e., OH and OI) emission layers.

In Fig. 9, the reconstructed time series of GW Event no. 01 and Event no. 02 are presented. In Fig. 9a, the reconstructed time series of the GW event on 4 December 2004 is shown. Subpanels (i) and (ii) show the reconstructed GWs of $\tau_1 = 25.47 \pm 02.40$ min and $\tau_2 = 33.47 \pm 03.16$ min, respectively. Based on the estimated phase difference ($\Delta\phi$) of 00.01 min of $\tau_1$, the vertical velocity ($V_z$) and wavelength ($\lambda_z$) go to infinity. This indicates very little or no phase difference. For $\tau_2$ of the GW event on 4–5 December 2004, OH leads OI by $\Delta\phi = 06.58$ min. This phase difference led to $V_z = 05.28 \pm 01.25$ m s$^{-1}$ and $\lambda_z = 10.60 \pm 02.50$ km. A summary of the estimated $\Delta\phi$, $V_z$, and $\lambda_z$ for Event no. 01 and Event no. 02 is presented in Table 3.

#### 5.1.2 Event no. 02

The phase propagation characteristics of GW Event no. 02 (1 May 2005) are shown in Fig. 9b. The individual reconstructed GWs for each period were constructed using the observation time of the data. In the subsequent subsections, the characteristics of the phases are discussed.

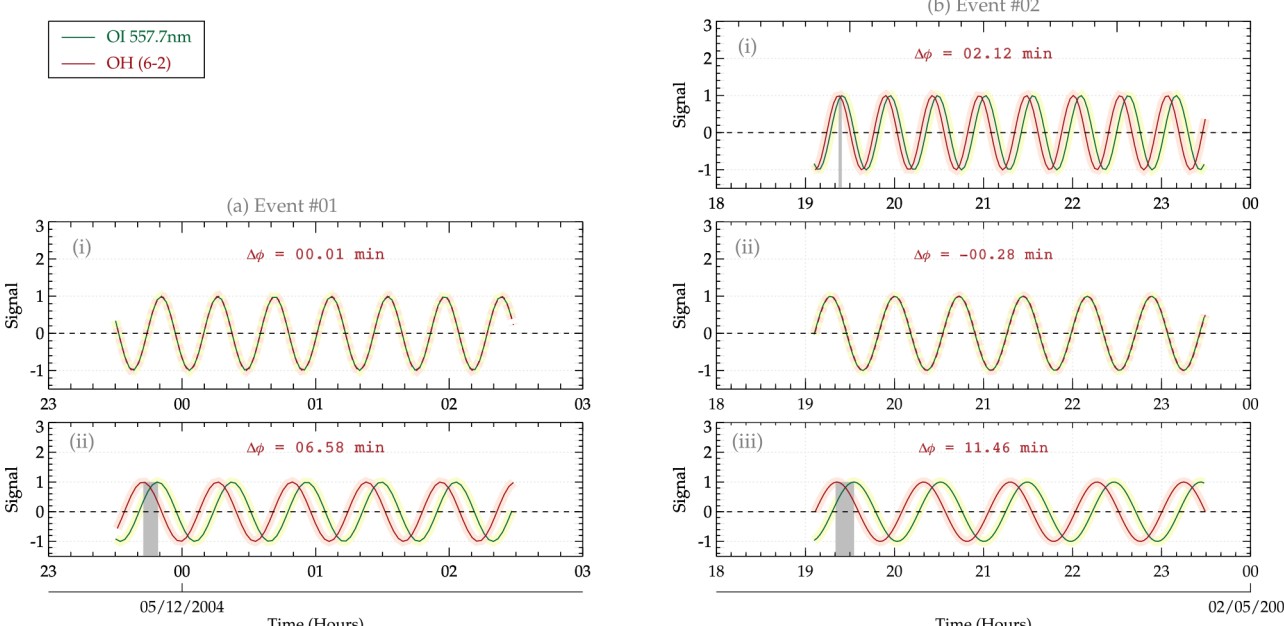

**Figure 9.** Two observed vertically propagating GW events in São João do Cariri. In panel **(a)**, the phase difference is determined using the reconstructed signal of OI and OH for Event no. 01. Using the reconstructed signal for each period in Event no. 02, the phase differences between OI and OH are determined and presented in panel **(b)**.

### $\tau_z = 31.64$ min

For Event no. 02 (see Fig. 6), propagation of the GW with $\tau = 31.64$ min shows a steep vertical downward phase propagation with altitude. Here, OH leads NaD by 0.698 min ($\sim 41.890$ s), NaD leads $O_2$ by 01.926 min, and $O_2$ lags OI by 00.502 min ($\sim 30.136$ s). In general, OH leads OI by 00.0354 h (02.122 min). In comparison, as shown in panel (i) in Fig. 9b, a phase lead was observed between the OH (solid red line) and OI (solid green line) emission layers. In panel (i) in Fig. 9, the phase difference ($\Delta\phi$) between OH and OI is represented by a positive (+) value to indicate a lead.

### $\tau_z = 43.25$ min

The second period ($\tau_z = 43.25$ min) of Event no. 02 demonstrated a mixture of phase leads and lags in the phase propagation from the OH through NaD and from the $O_2$ to the OI emission layers, showing steep vertical upward phase propagation for the first 4 h of the time series. No phase difference exists between OH and NaD, whereas NaD was found to lead $O_2$ by 0.00854 h ($\sim 00.512$ min/$\sim 30.748$ s). In the case of $O_2$ and OI, $O_2$ lags behind OI by 00.0132 h ($\sim 00.793$ min/$\sim 47.556$ s). Between OH and OI, OH lags OI by 0.00467 h ($\sim 0.280$ min/$\sim 16.808$ s). The phase propagation characteristics of this GW are shown in subpanel (ii) of Fig. 9b. Here, the $\Delta\phi$ is negative due to the phase lag.

### $\tau_z = 58.43$ min

The GW of period $\tau_z = 58.43$ min of Event no. 02 demonstrated a consistent phase lead from OH through NaD and from the $O_2$ to the OI emission layers, indicating vertical upward phase propagation between the four emission layers. The GW of this period indicates that OH leads NaD by 0.086 h ($\sim 05.136$ min), NaD leads $O_2$ by 0.0828 h ($\sim 04.970$ min), and $O_2$ leads OI by 0.0225 h ($\sim 01.351$ min). Generally, OH leads OI by 00.190 h ($\sim 11.456$ min). The phase propagation characteristics using the reconstructed time series of this GW are presented in subpanel (iii) of Fig. 9b. The $\Delta\phi$ is positive due to the phase lead.

GW phase propagation is used to determine the energy propagation (Nyassor et al., 2018, and references therein). A downward phase propagation indicates upward energy propagation and vice versa. In the case of Event no. 01, the phase across the four emission layers is upward and almost vertical; thus, the wave energy propagates almost vertically downward for the period ($\tau_1 = 25.47 \pm 02.40$ min). The characteristics of both of these events are similar, except that one of the periods of the GW in each event (i.e., $\tau_2$ of Event no. 01 and $\tau_3$ of Event no. 03) presented a well-defined vertical and upward phase propagation. The remaining period (as mentioned earlier) showed a very steep vertical phase line between the emission layers due to the small or nonexistent phase difference, which causes $V_z$ and $\lambda_z$ to approach infinity (presented in Table 3).

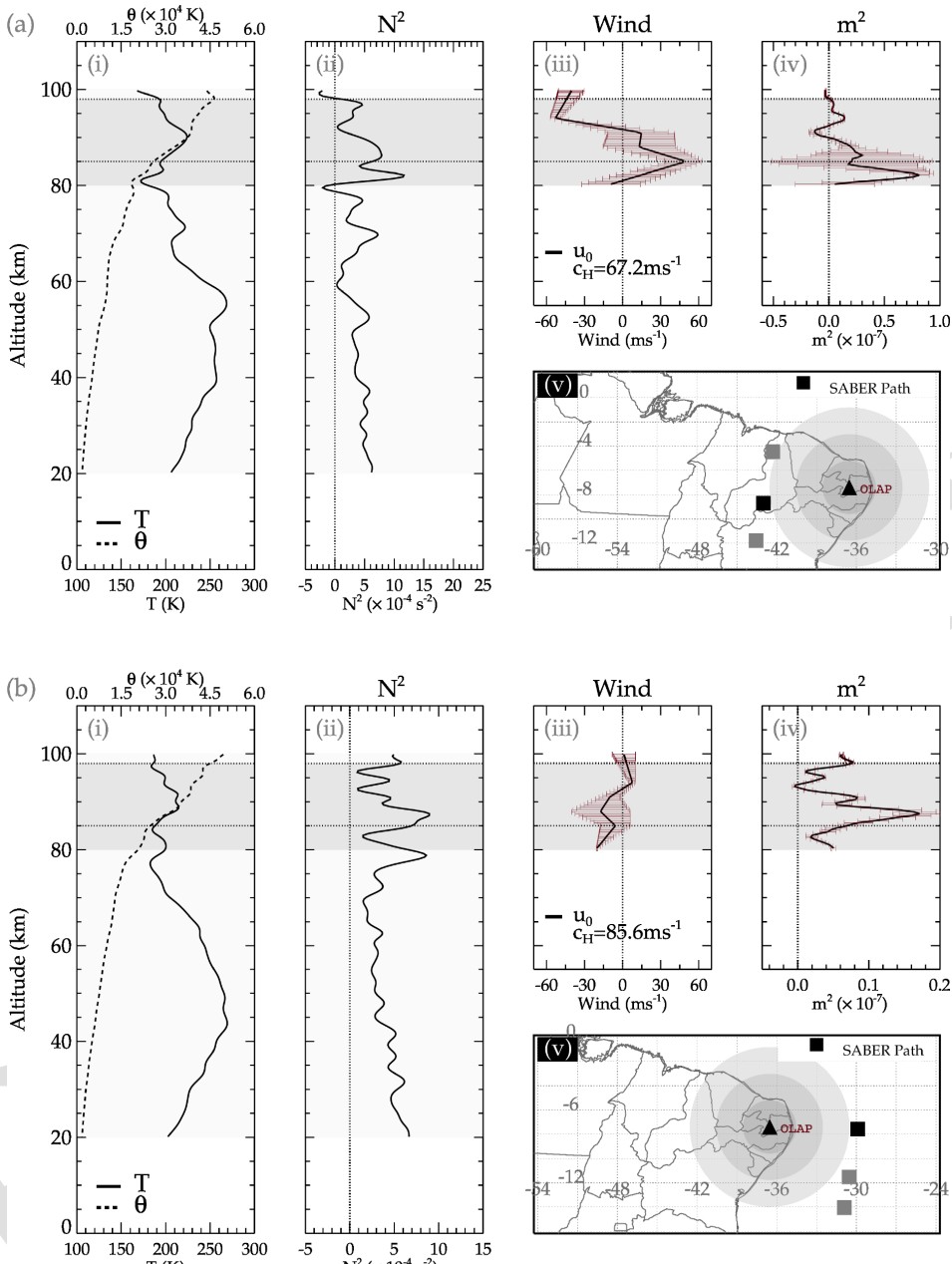

**Figure 10.** Propagation characteristics of the GW events observed on 4–5 December 2004 **(a)** and 1–2 May 2005 **(b)**. The kinetic temperature profile of the selected SABER sounding position of the profile in subpanel (i) is shown as a black square in subpanel (v). The dashed line in subpanel (i) is the profile of the potential temperature. In subpanel (ii), the profile of the Brunt–Väisälä frequency is presented. In subpanels (iii) and (iv), the profile of the wind in the direction of the GW propagation and the square of the vertical wavenumber ($m^2$) are presented, respectively.

Downward phase propagation indicates that this wave is generated upward and propagates downward (Vadas et al., 2018). Downward-propagating GWs, just like upward-propagating waves, transport momentum and energy from the source location and deposit this momentum and energy wherever they break or dissipate. However, these two cases presented three different dynamics:

1. almost vertical phase propagation across all emission layers;

2. an upward and downward phase propagation ("fish-bone" structure) at the later hours of the observation times; and

3. out-of-phase variation in the $O_2$ and OH rotational temperatures at the later hours of the observation time.

These dynamics, the propagation conditions of the GWs across these emission layers, and the characteristics of the momentum flux and potential energy are discussed in detail.

## 5.2 Background propagation conditions

Due to the characteristics of the phase propagation of the two events considered in this study, the propagation conditions between 80 and 100 km were investigated. Figure 10a TS2 and b represent the propagation conditions of GW Event no. 01 and Event no. 02, respectively. In subpanel (i) in Fig. 10a and b, the temperature profile obtained from the SABER observation (solid line) and the estimated potential temperature (dashed lines) are presented. The potential temperature ($\theta$) was estimated using Eq. (6). The profile of the Brunt–Väisälä frequency ($N$), presented in subpanel (ii) in Fig. 10a and b, was estimated using Eq. (5). The Brunt–Väisälä frequency profile is mostly used to examine the formation of ducts, also referred to as thermal ducts, due to the temperature gradient (Bageston et al., 2011). Ducts are known to trap vertically propagating GWs, causing them to propagate horizontally for longer distances and periods of time.

Doppler ducts, on the other hand, are caused by the background wind gradient (Bageston et al., 2011; Isler et al., 1997). To determine whether or not the propagation of GWs is hindered or favored by the background conditions (controlled mainly by wind and temperature), the square of the vertical wavenumber ($m^2$) profile is used. The $m^2$ can be estimated using the following expression (Bageston et al., 2011):

$$m^2 = \left[ \frac{N^2}{(u_0 - c_H)^2} - \frac{u_o''}{u_0 - c_H} - k_H^2 \right], \tag{10}$$

where $N$ is the Brunt–Väisälä frequency, $u_0$ is the observed horizontal wind in the direction of wave propagation, $u_o''$ is the second derivative of the wind with altitude, $c_H$ is the observed horizontal phase speed, and $k_H = 2\pi/\lambda_H$ is the horizontal wavenumber (with $\lambda_H$ being the horizontal wavelength). Equation (10) is a valid dispersion relation for GWs propagating in an environment in which the effects of horizontal wind and temperature gradient cannot be neglected (Chimonas and Hines, 1986).

The profile of the horizontal wind in the direction of the wave propagation, $u_o = u \cos\psi + v \sin\psi$, is shown in subpanel (iii) in Fig. 10a and b. Here, $u$ and $v$ are the respective zonal and meridional wind components and $\Phi$ TS3 is the propagation direction of the GW observed in the all-sky images. Using the other parameters defined in Eq. (10), the $m^2$ profile is estimated and plotted in subpanel (iv) in Fig. 10a and b.

In subpanel (v) in Fig. 10a and b, a geographical map of the OLAP facility and the sounding positions of the SABER satellite are presented. In this subpanel, the positions of the SABER soundings are indicated by squares, with the black square representing the temperature profile selected to compute the potential temperature and the Brunt–Väisälä frequency. The sounding positions during Event no. 01 began at $-11.77°$, $-43.54°$ CE12 at 00:47:29 UT; traveled through to 08.70°, $-43.00°$ at 00:48:30 UT; and continued on to $-04.45°$, $-42.28°$ at 00:49:41 UT. For Event no. 02, the sounding position began at $-07.56°$, $-29.90°$ at 18:46:35 UT; traveled through to $-11.51°$, $-30.54°$ at 18:45:24 UT; and continued on to $-14.04°$, $-30.90°$ at 18:44:39 UT. These sounding positions and times fall within a defined radius of 800 km around the observation site. This radius was defined so that any sounding that fell within it would be considered.

Several studies have demonstrated how the background conditions control the propagation characteristics of GWs. Using the $m^2$, the evanescent ($m^2 < 0$) and propagating ($m^2 > 0$) regions can be determined. Most ducts are formed when two evanescent regions exist above and below a region of $m^2 > 0$. In subpanel (iv) in Fig. 10a and b, the profiles of $m^2$ estimated using observed parameters are shown. As mentioned earlier, ducts are formed when there is a gradient in the background temperature and wind. In both cases, a gradient can be observed in the temperature and wind (see the gray shaded regions in subpanels i and iii), indicating the possibility of duct formation.

However CE13, the $m^2$ profile showed structures indicating the existence of a duct. During GW Event no. 01, two ducted regions can be observed: the first one between $\sim 91$ and 98 km and the second between $\sim 80$ and 90 km. The first duct most likely encloses the peak altitudes of the $O_2$ and OI emission layers. Even though this duct may appear quite weak, it is possible that it contributed to the trapping of the GW within this emission layer. The second duct, which is stronger than the first, most likely favored the trapping of the GWs at the peak altitudes of the OH and NaD emission layers. Despite the presence of the structure of a duct, the GW phase propagation presented a characteristic that appeared to be the effect of a single duct. This is because, except for a slight difference in the observed period of the $\tau_2$ GWs of Event no. 01 in the OH emission layer, all other emissions have the same period.

For Event no. 02, the structure of the duct presented almost the same characteristics as Event no. 01. An inversion layer could be observed in the temperature profile, as shown in subpanel (i) in Fig. 10b. However, this inversion layer did not significantly influence the formation of the duct, as shown subpanel (ii) in Fig. 10b. Only one observed evanescent region was formed at $\sim 94$ km. Around 83 km, a lower evanescent region can be seen forming, despite not attaining $m^2 < 0$. Even though the $m^2$ profile had the structure of a duct, only one evanescent region was formed. Besides this, the characteristics of the GWs observed in this event are similar to those of Event no. 01. Considering the propagation dy-

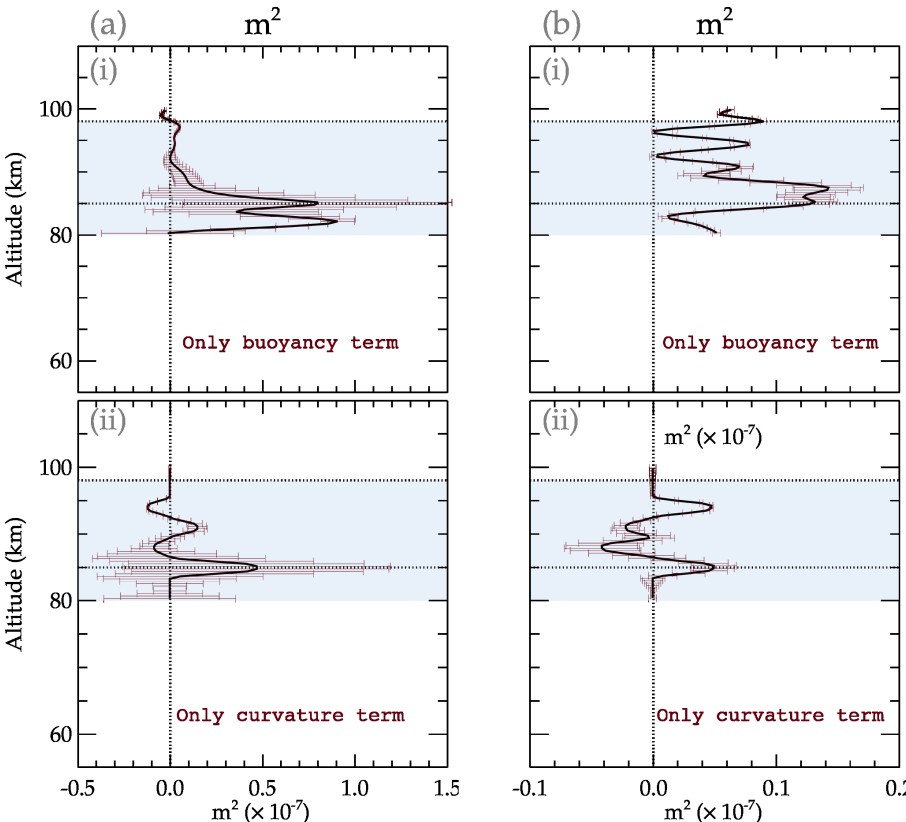

**Figure 11.** Assessing the influence of the buoyancy and curvature terms on the propagation characteristics of the GW events observed on 4–5 December 2004 **(a)** and 1–2 May 2005 **(b)**.

namics of this wave, which is suggestive of the influence of ducts, it is very probable that this structure might have played an important role in the dynamics of the vertical propagation of this event.

## 5.3   Assessing the duct formation and contributing factors

The formation of a duct and its dynamics are mainly controlled by wind and temperature. These parameters are defined in Eq. (10) and comprise the following: (1) the buoyancy term $(N^2/(u_0 - c_H)^2)$ resulting from the temperature and (2) the curvature term $(u''/(u_0 - c_H))$ due to wind. To carry out this assessment, either the buoyancy term or the curvature term is ignored in Eq. (10). Neglecting the curvature term (subpanel i in Fig. 11a), a broad duct extending from $\sim 80$ to 98 km is formed, with a maximum $m^2$ of $\sim 0.85 \times 10^{-7}\,\mathrm{m}^2$. However, the lower part of the evanescent region is not well formed, which can be attributed to the limitation of the wind data to 80 km. Below 93 km, the duct broadens to the lower (quasi-)evanescent region, extending down to 80 km. For Event no. 01 and considering only the curvature term (see subpanel ii in Fig. 11a), two weak ducts were formed with their centers at around 85 and 92 km with

a maximum $m^2$ of $0.5 \times 10^{-7}\,\mathrm{m}^2$ and $0.2 \times 10^{-7}\,\mathrm{m}^2$, respectively.

The assessment for Event no. 02 is presented in Fig. 11b. For the case in which the curvature term was ignored (considering only the buoyancy term), a duct with two maxima was formed between $\sim 80$ and 93 km with a peak of $\sim 0.16 \times 10^{-7}\,\mathrm{m}^2$. Above $\sim 93$ km, a narrow peak was also formed. Similar to the buoyancy term of Event no. 01, the lower limit of the evanescent region of this duct was not completely formed. Ignoring the buoyancy term (as shown in subpanel ii in Fig. 11a), the peak magnitude of the duct due to the curvature term is $0.04 \times 10^{-7}\,\mathrm{m}^2$ (see subpanel ii in Fig. 11b). Two weak ducts were formed with peaks at around 85 and $\sim 93$ km, with a broad evanescent region extending from $\sim 87$ to $\sim 92$ km.

From Figs. 10 and 11, it can be observed that there is the possibility of the existence of ducts. These ducts are mainly due to temperature gradients. However, the ducts are considered weak (due to low $m^2$ values), and multiple ducts may exist, with some not having a well-formed evanescent region ($m^2 < 0$). The observed ducts in this work can be considered not to comprise the entire structural dynamics, as the sounding positions during the GW events were quite distant from the observation site. From the phase lines, little or no phase

differences were observed during the first 2 h of Event no. 01 and approximately the first 3 h of Event no. 02. Using the phase difference, except for a period in each event, the estimated vertical wavelength ($\lambda_z$) approaches infinity and, thus, a vertical wavenumber ($m \to 0$). This characteristic causes the wave to undergo total internal reflection (Gossard and Hooke, 1975). Due to the limited altitude range of the photometer airglow observations, the full extent of the dynamics cannot be explored. This will be explored in a future paper.

## 5.4 Fishbone structure and out-of-phase temperature variation

Figures 7 and 8 show an upward and a downward phase progression, forming a fishbone structure (as described in the work of Vadas et al., 2018). In their work, the authors applied a procedure to explore whether or not the turning points in both cases (which occurred around the NaD emission layer) were a reflection point or a region in which dissipation occurred and whether these indicated possible excitation of other GWs' spectra. It is known that for GWs to be considered non-primary waves, as shown by various works (e.g., Vadas et al., 2003, 2018; Kogure et al., 2020; Heale et al., 2020), the GW propagation characteristics for the upward and downward components must meet certain criteria, as defined in Vadas et al. (2018). However, these criteria cannot be applied in this study due to the limited altitude range. Moreover, the events considered in this work present GWs with similar or nearly the same periods propagating through the NaD, $O_2$, and OI emission layers except for OH, which differs slightly but fell within the error margin. This shift in the period (see Fig. 5) could be due to the wind in the GW propagation direction.

Despite no existing evidence in the $m^2$ profile to explain the fishbone structure, the rotational temperature variation in the OH layer showed a shift and eventually became out of phase with the $O_2$ rotational temperature and the remaining emission layers (see Fig. 4). This characteristic was attributed to the response of temperature to the passage of GWs (e.g., Takahashi et al., 1990) and, probably, to reflected GWs CE14. Therefore, considering the fact that the OH rotational temperature is out of phase with the $O_2$ rotational temperature and the remaining emission intensity variation at the later observation times or usable window of the data considered in this work, the possibility of a reflection is indicated.

## 6 Momentum flux and GW potential energy

GWs transport momentum and energy from their excitation or source location to their sink (dissipation or breaking), regardless of whether the waves are propagating upward or downward (Fritts and Alexander, 2003; Vadas et al., 2009; Nyassor et al., 2021). The amplitude of upward-propagating GWs grows due to the decreasing density with increasing altitude. Therefore, for a downward-propagating wave, the

amplitude of the wave may suffer from an amplitude decrease due to increasing density with decreasing altitude. In Sect. 5.1, two GW events are selected with an upward phase propagation. The phases analyzed in Fig. 6 further showed phase leads and also small phase shifts. The individual reconstructed signal using the wave phases and periods affirms that a phase differences exists between the OI and OH emission layers.

According to Vadas (2007), diffusion processes inhibit the propagation of GWs where molecular viscosity and thermal diffusivity are significant in the upper mesosphere and lower thermosphere. Turbulent diffusion is also known to be a significant process that inhibits GW propagation in the lower and middle atmosphere (Yiğit and Medvedev, 2016). However, high-frequency GWs with relatively large intrinsic horizontal phase speeds mostly survive these conditions and are capable of directly propagating to the upper atmosphere, where they break or dissipate (Yiğit et al., 2021). Nevertheless, this poses the following question: what are the momentum flux and potential energy characteristics for either an upward- or downward-propagating GW when it happens to be in a duct? These features are evaluated using the momentum flux and potential energy of the GWs derived from the rotational temperatures of the $O_2$ and OH emission layers.

## 6.1 GW Event no. 01

The momentum and potential energy variation with time (and their averages) for the $O_2$ and OH emission layers are presented in Fig. 12 for the GW event on 4 December 2004 (Event no. 01). Note that GWs with the same or a similar period, determined using the variation in the intensity of the four emission layers, were determined in the rotational temperatures of the $O_2$ and OH layers. In Fig. 12a–d, the background temperature, residual, potential temperature, and Brunt–Väisälä frequency time series are presented, respectively. The solid black lines represent the $O_2$ time series, whereas the red line indicates the OH time series. The estimated momentum flux at the peak altitudes of the $O_2$ ($\sim 92$ km) emission layer and the OH ($\sim 87$ km) emission layer is shown in Fig. 12e. The momentum flux of each period (labeled $\tau_1 = 25.47$ min and $\tau_2 = 33.47$ min) of the GW in Event no. 01 is presented in subpanels (i) and (ii) of Fig. 12e. Subpanel (i), with the light-yellow background, highlights the period with steep vertical phase propagation. The temporal averages of the momentum fluxes for $O_2$ and OH indicate that the momentum flux of the GW in the $O_2$ layer ($M_{F(O_2)}$) is greater than the momentum flux in the OH emission layer ($M_{F(OH)}$) for $\tau_2$. However, for $\tau_1$, the momentum flux in both emission layers is approximately 0 ($M_{F(O_2)}$ $\sim M_{F(OH)} \sim 0$). The potential energy ($E_p$), on the other hand, is higher in the $O_2$ layer compared with that in the OH layer.

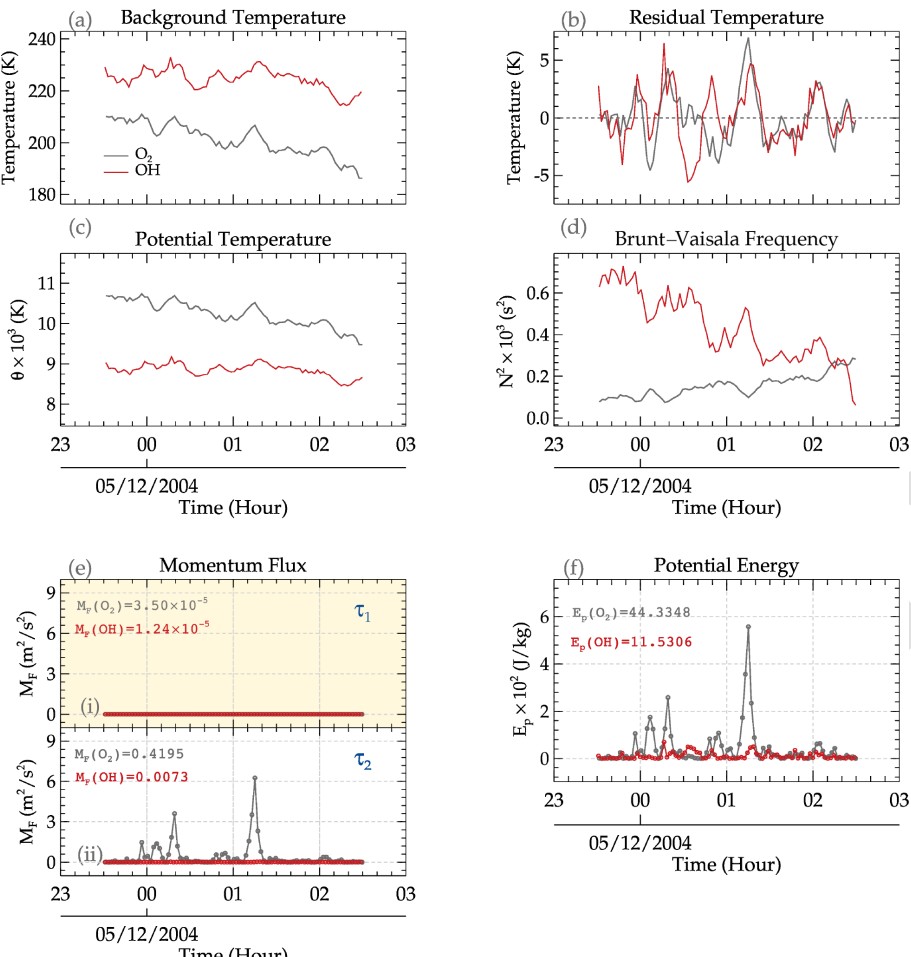

**Figure 12.** The characteristics of momentum flux **(e)** and potential energy **(f)** at the $O_2$ and OH emission altitudes for the event on 4 December 2004. In subpanels (i) and (ii) of panel **(e)**, the GW momentum flux for first and second period of Event no. 01 are presented. The light-yellow background (in subpanel i) highlights the period with steep vertical phase propagation. In panels **(a–d)**, the background temperature, residual temperature, potential temperature, and Brunt–Väisälä frequency time series are presented, respectively.

## 6.2    GW Event no. 02

Similar to Event no. 01, the background temperature, residual temperature, potential temperature, and Brunt–Väisälä frequency time series are presented in Fig. 13a–d, respectively. The momentum flux and potential energy at each emission layer for the three GWs observed during the 1 May 2005 event are presented in Fig. 13e and f. The momentum flux of three periods (labeled $\tau_1 = 31.64$ min, $\tau_2 = 43.25$ min, and $\tau_3 = 58.43$ min) of the GW in Event no. 02 is presented in subpanels (i), (ii), and (iii) in Fig. 13e. The periods with steep, near-vertical phase propagation (i.e., $\tau_1$ and $\tau_2$) are shown in subpanels (i) and (ii) using a light-yellow background. The temporal averages of the momentum fluxes for $O_2$ and OH indicate that the momentum flux of the GW in the $O_2$ layer ($M_{F(O_2)}$) is greater than the momentum flux in the OH emission layer ($_{F(OH)}$) for all three periods determined in this event. It is observed that the amplitude of the mo-

mentum fluxes for each emission layer increases with period (thus, $M_{F_{\tau_1}} < M_{F_{\tau_2}} < M_{F_{\tau_3}}$).

Comparison between the momentum fluxes at the $O_2$ and OH emission layers for this event showed a vast difference. The $M_F$ at the $O_2$ layer is much higher than that at the OH layer. This difference can be attributed to the large amplitude of the GW perturbations in the $O_2$ temperature residual, as the estimations of the potential energy and the momentum flux depend on the temperature residual. However, for the GW event on 4–5 December 2004, the amplitudes of the temperature perturbations due to GWs are similar. The factor that made the momentum flux greater in the $O_2$ altitude was the spikes. Besides these spikes, the momentum flux at the $O_2$ and OH altitudes is similar. Another explanation for this difference, apart from the increase in amplitude due to a decrease in density with altitude, is $N^2$. The time series of $N^2$ varied considerably over the observation window, especially for the OH emission layer, for both events.

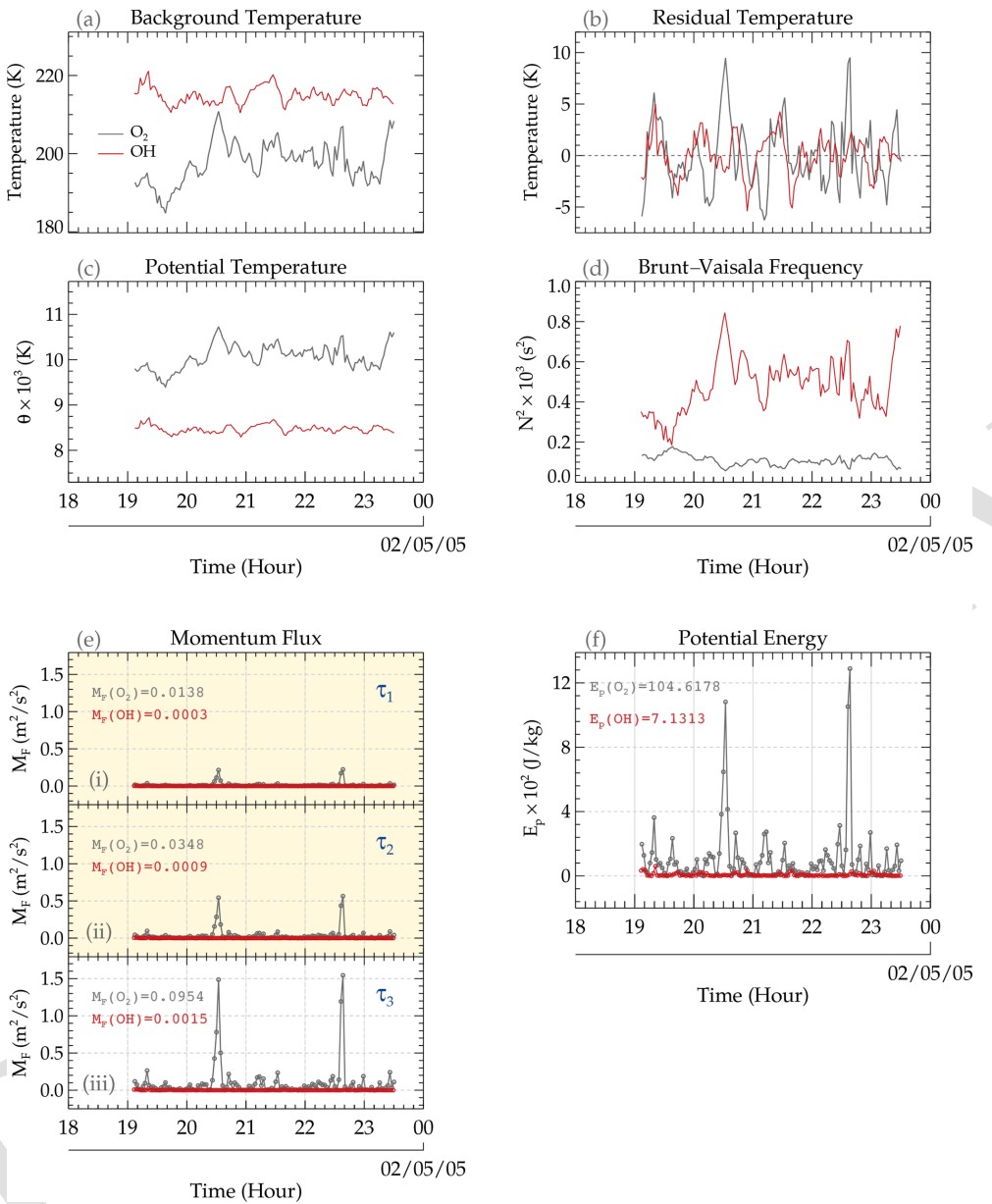

**Figure 13.** The characteristics of the momentum flux **(e)** and potential energy **(f)** at the $O_2$ and OH emission altitudes for the event on 1 May 2005. In subpanels (i), (ii), and (iii) of panel **(e)**, the GW momentum fluxes for the first, second, and third periods of Event no. 02 are presented. The light-yellow background (in subpanels i and ii) highlights the periods with steep vertical phase propagation. In panels **(a)**–**(d)**, the background temperature, residual temperature, potential temperature, and Brunt–Väisälä frequency time series are presented, respectively.

Comparison between the two events showed that, regardless of the vertical propagation of GWs, momentum and energy are transported from the source to the sink. It is imperative to say that the momentum and energy at the source will be lower. This has clearly been demonstrated using the two selected events, attesting to the fact that atmospheric density significantly impacts the amplitude, momentum, and energy. Using a lidar temperature profile, Kaifler et al. (2017) studied the dynamics of downward-propagating GWs. They observed that one-third of the momentum flux is carried by the downward-propagating GW from an altitude of 85 km to a lower altitude. In this work, this characteristic could not be accounted for due to the presence of ducts that could possibly change the dynamics of momentum and energy transport. In general, the momentum flux and potential temperature for both events are higher in the $O_2$ emission layer than in the OH layer. It can be postulated that if a duct indeed exists in each event, the momentum and potential energy should be higher in the peak region of the duct, where $O_2$ is situated. At the reflection points (where $m^2 < 0$), the momentum

flux and the potential energy should be lower. This postulate could not be explored due to a lack of high-resolution vertical observations between 80 and 100 km. This subject is, however, something that the authors intend to examine in a future paper.

The momentum fluxes of the GWs of periods with steep vertical propagation, i.e., $\tau_1$ of Event no. 01 and $\tau_1$ and $\tau_2$ of Event no. 02, presented different characteristics. As mentioned earlier, the estimated momentum flux of $\tau_1$ of Event no. 01 in both emission layers is near zero, indicating no exchange of momentum between the two layers. The near-zero momentum flux corresponds to $\tau_1$, which has steep vertical phase propagation. The phase propagation of the GW with this period is almost the same for each emission, which caused the $V_z$ and $\lambda_z$ to approach infinity. For $\tau_1$ and $\tau_2$ in Event no. 02, even though the momentum flux in the $O_2$ emission layer is higher than that in the OH layer, the values are lower compared with $\tau_3$. Their phase propagation is almost vertical, although not as in the case of $\tau_1$ for Event no. 01.

Considering the fact that the vertical propagation of these GWs is only within a 5 km range, the full extent of their characteristics cannot be explored due to the unavailability of data. The potential energy of Event no. 01 also depicted characteristics similar to the momentum flux. A lower momentum flux and potential energy at the OH emission layer is indicative of no deposition of momentum and energy, whereas higher values in the $O_2$ layer point to deposition. This illustrates the governing theory of the transport of momentum and energy by atmospheric GWs. Using a longer altitude range and high-temporal-resolution lidar data, for instance, this subject can be explored in detail, and standards can be defined to determine the signatures of vertically propagating GWs due to reflection and non-primary GWs. In summary, this study has illustrated the possible consequences of ducts on the vertical propagation of GWs and their dynamics (specifically, momentum and energy transport).

## 7   Conclusions

This paper studies the dynamic characteristics of the momentum flux and potential energy of vertically propagating (almost downward-propagating) GWs using two GW events selected for case studies. Using the phase propagation of GWs with almost the same periods through the emission layers of OI 5577, $O_2$, NaD, and OH, the vertical propagation of the waves was determined. Using the ratio of the altitude difference ($\Delta d$) to the phase difference ($\Delta\phi$), the vertical phase speed and, consequently, the vertical wavelength were estimated. From the phase propagation, two classical events with almost downward-propagating GWs were selected for further study.

The potential energy and momentum flux were estimated, and their characteristics were studied. For each propagating GW, it was determined that the momentum flux and potential energy at the $O_2$ emission altitudes were higher than those at OH emission altitudes. The momentum flux and the potential energy at the OH emission altitude were far lower than that of the momentum flux at the $O_2$ emission layer. No distinct amplitude enhancements of GWs were observed in the $O_2$ layer compared with GWs observed in the OH emission layer. The similar GW amplitude of propagating waves in the $O_2$ and OH rotational temperatures indicates a restricted or bounded propagation condition.

Due to the steep phase difference between the four emission layers, background propagation conditions are investigated. The $m^2$ profile suggests the presence of ducts during the two events. These ducts were found to be created due to temperature inversion. However, there is no information about the full extent and dynamics of the ducts, as the SABER sounding positions from which the temperature profile was computed were at a distance from the observation sites. However, $m \to 0$, caused by a steep phase line, suggests total reflection, indicating the presence of ducts. The presence of a duct is further confirmed by the lack of an amplitude difference between the two emission layers.

Toward the end of the observation or time series, it was observed that the OH rotational temperature was out of phase with respect to the intensity variations in OI, $O_2$, NaD, and OH as well as the $O_2$ rotational temperature. This characteristic is caused by the reflection of GWs. The observed duct could not be used to support or explain why the OH rotational temperature was out of phase. The fishbone structure that formed at the end of the observation period is typically suggestive of reflection, as GWs with similar periods propagated across.

As mentioned earlier in this work, the altitudinal difference between the two rotational temperatures is limited to only 5 km (i.e., only in the mesopause region). Hence, to come to a definite conclusion as to whether or not the steep phase lines, high $\lambda_z$, and the same or similar GW amplitude (residual temperature) are due to the imposing propagation restriction by the duct, a detailed study using co-located observations of higher spatial (vertical) resolution is required. However, this work demonstrated that momentum and energy deposition are affected in the presence of a duct. This characteristic can aid in the setting of boundary conditions when considering the vertical propagation dynamics of GWs.

## Appendix A: Retrieval of gravity wave parameters from OH all-sky images and meteor radar wind dynamics

### A1   Spectral and keogram analysis

In order to extract the parameters of GWs, a discrete Fourier-transform-based spectral analysis was used. First, a region containing GW oscillations was selected in both the zonal and meridional components of the keogram components (as

shown in Fig. A1). Note that the same area in each of the components was considered for analysis. Next, a discrete Fourier transform (Eq. A1) was applied to the selected areas (Wrasse et al., 2007; Figueiredo et al., 2018).

$$F(\omega) = \sum_{n-0}^{N-1} f(t)e^{\frac{-2\pi \omega n_i}{N}} \tag{A1}$$

Here, $F(\omega)$ is the transform of the Fourier function $f(t)$; $\omega = 0, \ldots, N-1$ is the frequency index; and $N$ is the number of points in the time series within the selected regions. Then, the cross-spectrum defined as follows:

$$C(x) = F_s(\omega)F_{s+1}^*(\omega), \tag{A2}$$

where $C(\omega)$ is the cross-spectrum between two time series and $F_s(\omega)$; $F_{s+1}^*(\omega)$ represents the Fourier transform of the series $f_s(t)$ and $f_{s+1}(t)$, respectively; and $F_{s+1}^*(\omega)$ is the complex conjugate of $F_{s+1}(\omega)$. The one-dimensional cross-power spectrum is defined by the quadratic modulus, $|C2|$. The amplitude of the cross-power spectrum is then determined using $2\sqrt{|C^2|}$, with the phase of the cross-spectrum defined as follows:

$$\Delta\psi = \tan^{-1}\left\{\frac{\mathrm{Im}(C(\omega))}{Re(C(\omega))}\right\}, -\pi \le \psi \ge \pi. \tag{A3}$$

The phase difference between these time series caused by the wave propagation is considered to be the frequency $\omega$, corresponding to the maximum amplitude. From the above estimations, the wave parameters are determined as follows:

1. period (min),

$$\tau = \frac{1}{|f(\omega)|}; \tag{A4}$$

2. horizontal wavelength (km),

$$\lambda_{\mathrm{H}} = \frac{\lambda_{\mathrm{NS}}\lambda_{\mathrm{EW}}}{\sqrt{\lambda_{\mathrm{NS}} + \lambda_{\mathrm{EW}}}}. \tag{A5}$$

Here, the wavelength (in km) for the zonal and meridional components ($\lambda_{\mathrm{NS}}, \lambda_{\mathrm{EW}}$) is $\lambda_{\mathrm{NS,EW}} = \frac{\Delta d}{\Delta\psi/360°}$, in which $\Delta d$ is the distance between the time series.

3. The horizontal phase velocity $c_{\mathrm{H}}$ ($\mathrm{m\,s}^{-1}$), and phase propagation direction $\phi(°)$, are determined by

$$c_{\mathrm{H}} = \frac{\lambda_{\mathrm{H}}}{\tau} \text{ and } \phi = \cos^{-1}\left(\frac{\lambda_{\mathrm{H}}}{\lambda_{\mathrm{NS}}}\right). \tag{A6}$$

Five GWs were detected from these two events using the abovementioned spectral analysis. These results are presented below.

In the upper panels of Figs. A1, A2, A3, A4, and A5, the left side is the zonal keogram, whereas the right side is the meridional (merid) keogram. These keograms correspond to the selected region with GW perturbations. The lower panels of Figs. A1, A2, A3, A4, and A5 show the amplitude (left), with the GW parameters listed along with their corresponding standard deviation ($\sigma$) in the lower right-hand panel. The horizontal dotted red lines indicate a significance level greater than 95.0 %, whereas the red circle with a black dot shows the peak amplitude. The GW characteristics in the lower panels of Figs. A1, A2, A3, A4, and A5 are the horizontal parameters (i.e., the sum of the zonal and meridional components).

## A2 Observed horizontal wind during the gravity wave events

In Figs. A6 and A7, the zonal (panel a) and meridional (panel b) winds are presented.

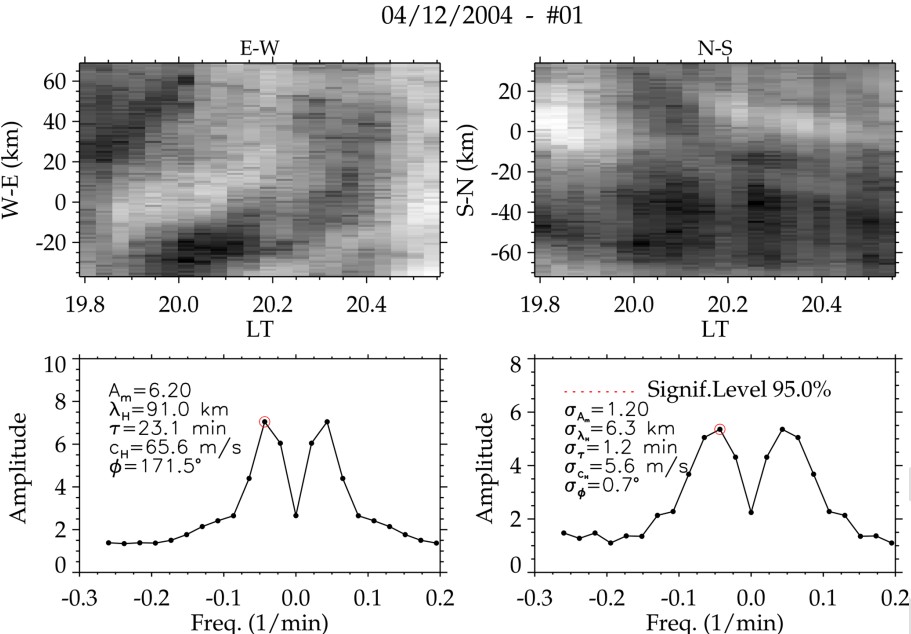

**Figure A1.** Results of the OH emission layer keogram and spectral analysis of Event no. 01 with a period of $\tau = 23.10 \pm 01.20$ min.

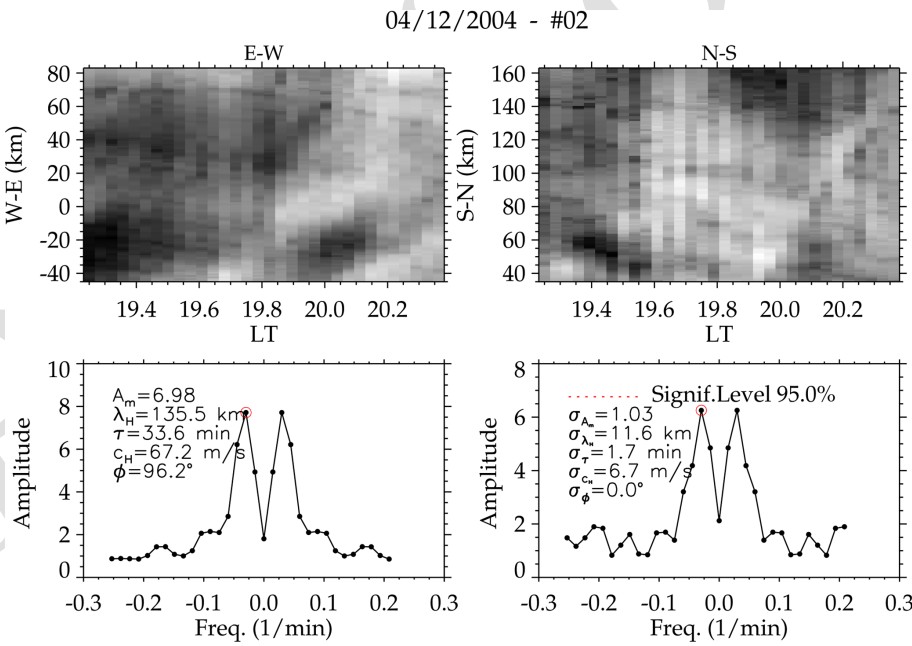

**Figure A2.** Results of the OH emission layer keogram and spectral analysis of Event no. 01 with a period of $\tau = 33.60 \pm 01.70$ min.

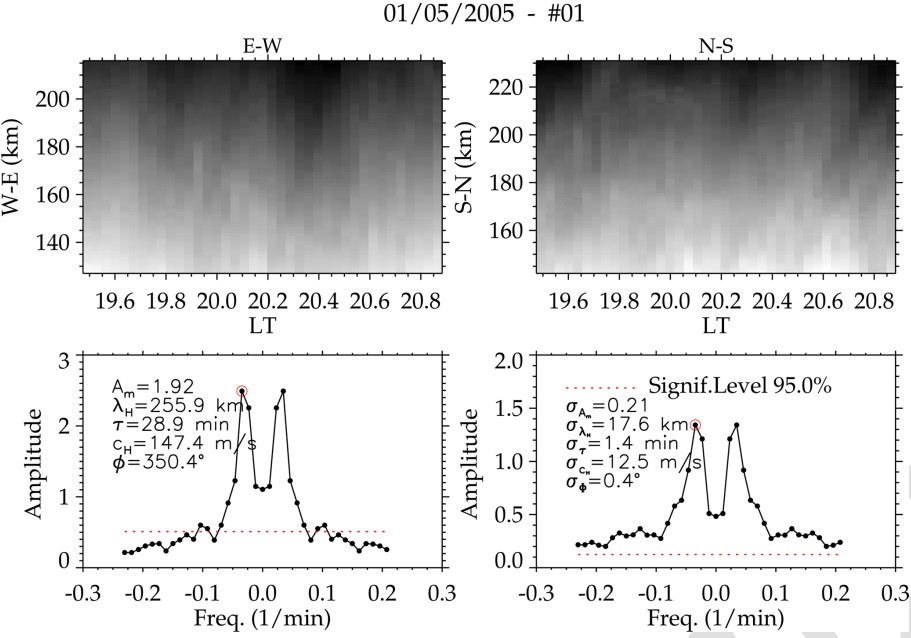

**Figure A3.** Results of the OH emission layer keogram and spectral analysis of Event no. 02 with a period of $\tau = 28.90 \pm 01.40$ min.

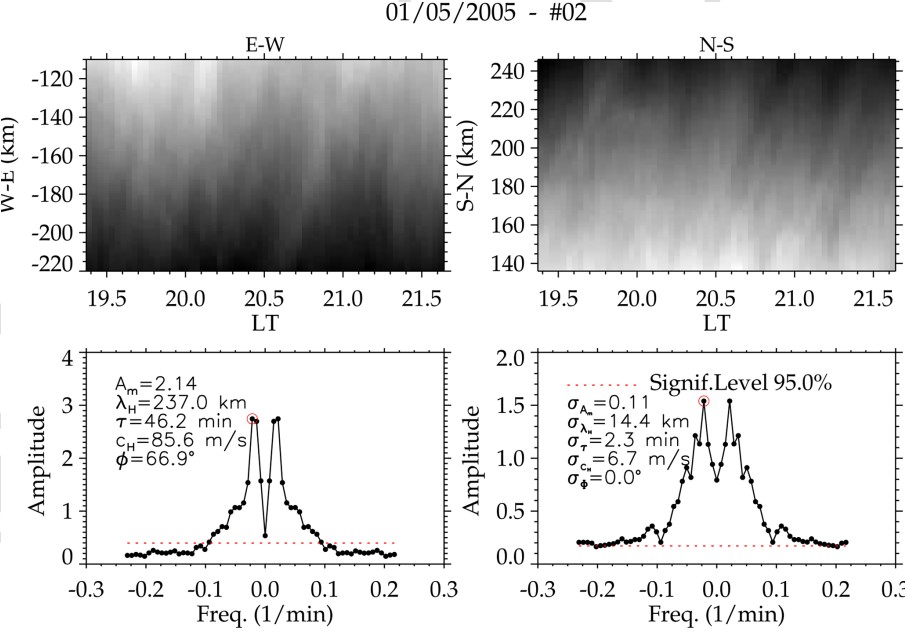

**Figure A4.** Results of the OH emission layer keogram and spectral analysis of Event no. 02 with a period of $\tau = 46.20 \pm 02.30$ min.

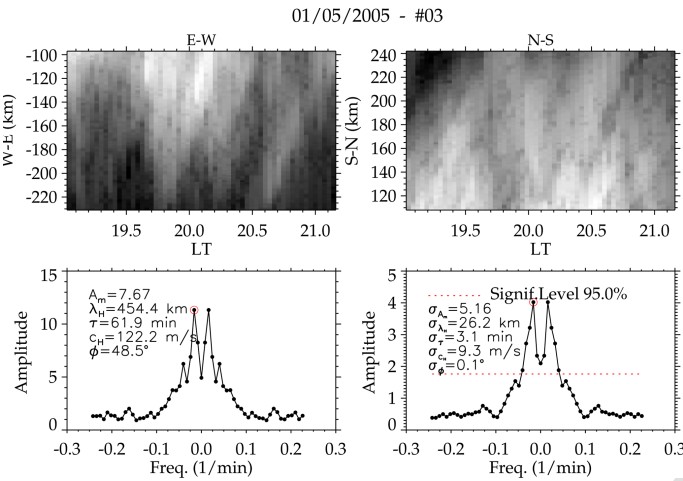

**Figure A5.** Results of the OH emission layer keogram and spectral analysis of Event no. 02 with a period of $\tau = 61.90 \pm 03.10$ min.

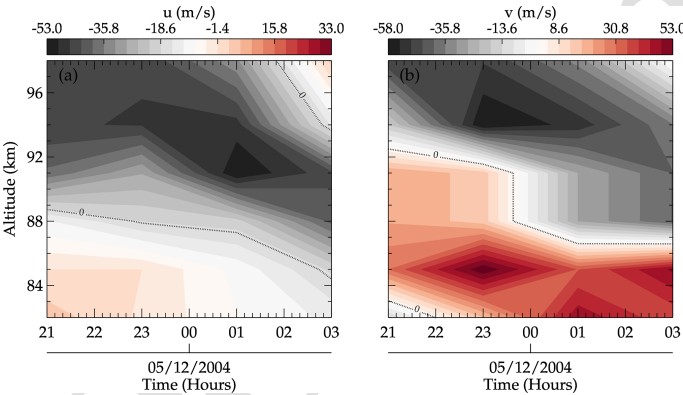

**Figure A6.** Meteor radar winds during the 4 December 2004 GW event at São João do Cariri. The zonal and meridional winds are presented in panels **(a)** and **(b)**, respectively.

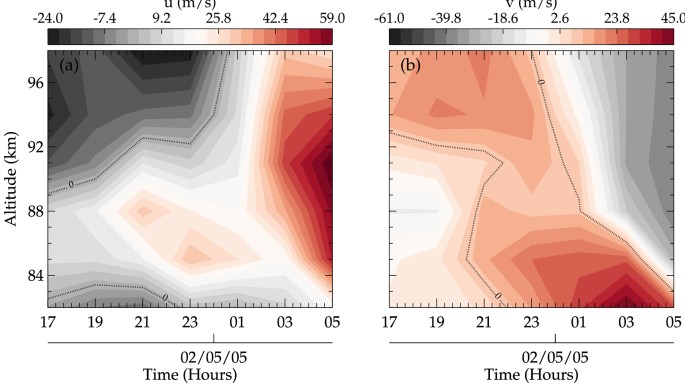

**Figure A7.** Same as Fig. A6 but for the 1 May 2005 GW event at São João do Cariri.

**Code availability.** The wavelet software was provided by Christopher Torrence and Gilbert P. Compo and is available at http://atoc.colorado.edu/research/wavelets/ (Torrence and Compo, 1998).CE15

**Data availability.** The photometer and meteor radar data used to generate the results in this paper were obtained from the Observatório de Luminescência Atmosférica da Paraíba at São João do Cariri, which is supported by the Universidade Federal de Campina Grande and Instituto Nacional de Pesquisas Espaciais. The data are available upon request from either Ricardo A. BuritiTS4 (rburiti@df.ufcg.edu.br) or Cristiano M. Wrasse (cristiano.wrasse@inpe.br). The airglow data are available from the "Estudo e Monitoramento Brasileiro do Clima Espacial" (EMBRACE/INPE): http://www2.inpe.br/climaespacial/portal/en (EMBRACE, 20024). The TIMED/SABER data are available from https://saber.gats-inc.com/data.php (SABER, 2024).

**Author contributions.** PKN wrote the article and performed most of the analysis. CMW and EY assisted with validation of the methodology and revision of the manuscript. IP, VYTB, CAOBF, and GAG assisted with validation of some of the methodology and with revision of the manuscript. RAB provided the photometer and meteor radar data and aided with revision of the manuscript. FE processed the meteor radar wind data and revised the manuscript. OMA, HT, and DG revised the manuscript.

**Competing interests.** The contact author has declared that none of the authors has any competing interests.

**Acknowledgements.** Prosper K. Nyassor and Vera Y. Tsali-Brown acknowledge support from the Fundação de Amparo à Pesquisa do Estado de São Paulo (FAPESP). The authors also thank the Coordination for the Improvement of Higher Education Personnel (CAPES) and the National Council for Scientific and Technological Development (CNPq) for support. Moreover, thanks are given to the Brazilian Ministry of Science, Technology and Innovation (MCTI) and the Brazilian Space Agency (AEB). We also extend our thanks to the Brazilian Ministry of Science, Technology, and Innovations (MCTI) and the Brazilian Space Agency (AEB). Igo Paulino, Gabriel A. Giongo, Oluwasegun M. Adebayo, and Hisao Takahashi thank CNPq for financial support. Igo Paulino, Cosme A. O. B. Figueiredo, and Fabío Egito acknowledge the Paraiba State Research Foundation (FAPESQ). The authors thank the Estudo e Monitoramento Brasileiro do Clima Espacial (EMBRACE/INPE), for the provision of the all-sky data, and the OLAP and UFCG, for the provision of the photometer and meteor radar

data. Finally, we acknowledge the Sounding of the Atmosphere using Broadband Emission Radiometry/Thermosphere Ionosphere Mesosphere Energetics Dynamics (SABER/TIMED) team for the temperature profiles.CE16

**Financial support.** This research has been supported by the Fundação de Amparo à Pesquisa do Estado de São Paulo (FAPESP; grant nos. 2021/07491-6 and 2018/10679-4); the Conselho Nacional de Desenvolvimento Científico e Tecnológico (CNPq; grant nos. 309981/2023-9, 140401/2021-0, 161901/2022-0, and 310927/2020-0); the Paraiba State Research Foundation (FAPESQ; grant nos. 2417/2023 and 3032/2021); the Brazilian Ministry of Science, Technology, and Innovations (MCTI) and the Brazilian Space Agency (AEB) (grant no. 20VB.0009); and the Coordination for the Improvement of Higher Education Personnel (CAPES; 001CE17).

**Review statement.** This paper was edited by John Plane and reviewed by three anonymous referees.

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

## Remarks from the language copy-editor

## Remarks from the typesetter