# Peer review of "Momentum flux characteristics of vertical propagating Gravity Waves"

_EGUsphere, 2024_

## Author Comment (AC1)

**Review for paper "Momentum Flux characteristics of vertical propagating Gravity Waves" by P. Nyassor et al.**

**General Comments**

The paper shows results from a case study of 2 mesospheric gravity wave events above São João do Cariri. They are able to demonstrate that the momentum flux differences between two different altitudes agrees with what is expected from theory with regards to upward and downward propagating waves. They show this in their figures and explanations in the text. Some refinement of the figures and explanations in the text are required, in my opinion, before it can be accepted for publication.

**Specific Comments**

This paper examines 2 nights of data and compares them. One of these nights the meteor radar data is not available, so they are not able to do a complete analysis comparison of the energy. Are these the only 2 nights with a similar period GW in all four filters over the 7 year dataset? Are there no others where the meteor radar is working so you can do a full comparison of momentum flux and energy?

**Response:**

So far, only two cases that have been with quite a good phase difference and similar periods were detected in four emission layers. However, we do have quite a number of cases of similar GWs detected in three emission layers. These cases were not selected due to the fact that we want to create a room for the detection of a fish bone structure (as it is in the case 04 – 05 December, 2004) in the phase propagation of the GWs.

With respect to the wind data from meteor radar, the data available was not processed and as at the time of submission of the manuscript. Also, since the initial objective was to investigate the dynamics of the momentum flux, we did not deemed it necessary whether we have wind data or not, hence, the omission of the estimation of the kinetic (total) energy.

Due to your comment and that of referee two, the wind data has been processed and the analysis has been performed on both momentum flux and energy for both cases.

Abstract – please mention that you are only looking at 2 events at the start of the abstract to aid clarity. You also mention reflected non-primary waves in the abstract, but surely this technique can just show that the wave observed is up/downward propagating, you can't say whether it is primary/non-primary or a reflected wave of any order?

**Response:**

Throughout – you use the term "energy" throughout the paper, but given the potential energy can only be examined for both events I would recommend altering this phrasing to reflect the results.

**Response:**

We appreciate your comment. In reference to your comment on the wind data for the case of 21 – 22 May, 2006, we have been able to obtain the data, thus the full analysis has now been realized, and hence a full but modified comparison has been done on the energy as well.

Section 2.1 – please include at least the altitudes of the different airglow layers at the start of this section to aid the reader or someone who is new to airglow studies. I know that you have pointed to the Nyassor et al paper, which does contain all the details, and mention them much later in the paper but the basics that are relevant to this study should be included early on.

This section has been rectified.

Section 2.2 – please include the height range that the radar observes at, you've mentioned the vertical resolution, but the height range is needed for context.

The altitude range of observation of the meteor radar has been included.

Line 124 – it is not clear what is meant by "19-25" hours. If the spikes in Fig. 2a are between 21-23 hours then the remaining dataset left is between 23 and 27, as per Fig. 2b-d, not 19-25 hours.

This has been corrected. In the new version, the time format has been changed. As a result, the sentence is "Due to the spike in the time series in Figure 3a, the data is limited to 23 UT on 04/05/2004 to 03 UT on 05/05/2004".

Section 3.3 – you mention Lomb-Scargle periodogram in this section but only show results from the wavelet analysis, is the L-S method used in this paper? If so, how closely does it match the wavelet results?

Thanks very much for the comments and questions. The periods present in the Lomb-Scargle are within the same range of the periods. Due to your question, Figure 5 and 6 had been included show the step by step analysis procedure and the Lomb-Scargle result.

Section 3.3 – the widths of some of the airglow layers overlap, does that influence your results at all?  Also, the altitudes given in this section (see my earlier comment about section 2.2) are the average altitudes for the layers, but there has been work that shows that these layers do tend to vary in their altitude over time, would this affect your interpretation of the results?

Thanks for the question. The width of the emission layers do overlap but will not influence the interpretation of the results. The condition for a vertical propagating gravity wave is that the vertical wavelengths must be larger than the airglow layer thickness (typical full width at half maxima, ~8 km) can be observed at multiple airglow emissions almost simultaneously. Secondary, since bandpass filters are used to select the emission layer of interest, the variation of the emission layer attitude over time will not affect the interpretation of the results. The only possible effect will be either an increase or a decrease in the intensity of the layer.

Figure 3c – this is not very clear, the dotted lines all look near vertical apart from the last one in the bottom two lines. Maybe this needs to be highlighted in the figure. Also, it needs to be clear which of the two observed gw periods this is referring too or if they're combined somehow. E.g. Fig 5 is much clearer.

In Figure 3c, the reconstructed signal includes the two peak periods. The plot has been enhanced to depict the phase change using the dashed lines. Due to the size of Figure 3c (now Figure 7c), the Figure 5 (now Figure 8c) was produce in the results section in order to make ensure the phase difference are well depicted.

Figure 3d – is this just for the airglow intensity (photometer data) or the temperature data?

Yes, the wavelet was only applied to the intensities only. This is because the phase change is estimated using the intensities. For the temperature, they are used for the estimation of the momentum flux and the potential energy. From Figures 5 and 6, both intensities and temperature were subjected to the Lomb-Scargle. This was done to verify that the observed GW modulated the temperature profile.

Figure 4: - Figures e-h are duplicated in Fig. 7 – do they need to be?

This was done to put emphasis on the dynamics of the momentum flux, potential, kinetic and total energy. However, due to the comment of the referee 2, the Figures regarding this parameter has been modified.

- 4 g and h need to have the same X-scale as the rest of the plots (same for Fig. 7) to help with interpretation.

  Well-noted. Your comment has been implemented in a modified version.

- 4e and f – these are on a different temporal resolution than KE, could you show them on the same scale but with error bars on to represent the small-scale variability seen in the MF and PE plots.

Thanks very much for the comment. Due to the comment of the referee 2, the Figures and the respective analysis have been modified.

Section 6.2, 300-301 – can the phrases " a small fraction" and "a great amount" be replaced with something more precise please.

Rewrite this section when discussion.

**Technical comments:**

Line 124 – I think you mean Figure 2a not 1a

Figure 1a has been replaced with 3a.

Line 272 – the word wave is missing at the end of this sentence.

The word "gravity wave" has been included at the end of the sentence in Line 510 in the new version of the manuscript.

Line 273 – replace "using" with "calculated from".

The word "**using**" has been be changed to "**calculated from**"

**Citation**: https://doi.org/10.5194/egusphere-2024-1982-RC1

---

## Author Comment (AC2)

**Response to Referee #02**

**Overview:**

This manuscript presents results of two gravity wave events at 36.31W, 07.40S using photometer, all-sky imager, and meteor radar data. The authors use this data to determine characteristics about the present waves and associated momentum and energy fluxes. The technique presented is interesting, and could provide beneficial scientific information. It is a useful idea from the authors to use multiple airglow layers to better understand gravity wave propagation in the MLT region. Nevertheless, there are several issues with the manuscript that are concerning. Importantly, the calculated values need to be better explained in the context of what assumptions were made, and what specific measurements were used. The "reconstructed" waves need more discussion and justification for how the wave parameters were chosen.

It is concerning that the kinetic energy calculation does not seem to be for the wave itself, rather it is based off of >1hr wind perturbation measurements (1hr resolution) that do not have the resolution to capture the waves being studied in this manuscript. Additionally, there is lacking information on exactly how the momentum flux was calculated. A temperature perturbation value is used, but it is not clear how this was obtained. It should be the average perturbation value over the wave packet, or the actual temperature perturbation amplitude. Instead, it appears that the raw residual temperatures were applied directly to the MF calculation.

Given these aspects, and the other concerns listed below, I am suggesting the paper be rejected. If these analysis issues can be mitigated, and the techniques more properly explained, there is potential that the manuscript could be resubmitted.

**Detailed concerns are listed below:**

1. **Lines 67-68 "The temporal resolution of the observation is 2 minutes, thus GWs with periods greater than 2 minutes can be observed." Lines 67-68**

The airglow photometer that measures OI, O2, NaD, and OH is said to have a 2 minute resolution. That would mean that GWs with periods of greater than 4 minutes can be observed (at best). More importantly, the authors need to provide explanation as to how phase shifts can be determined with a 2-minute resolution. Also, a 3-point running mean has been applied to the data (discussed on lines 125-126), which would further reduce the resolution. Furthermore, the data itself will have associated noise. The fit (equation 2) would also have some errors associated with it. So, how does this affect the calculation of phase differences between the different airglow layers?

**Response:**

As you have rightly said, the 3-point running mean further reduced the resolution. However, considering your comments, a 5-point running mean has rather been applied to obtain a resolution of 10 minutes. With his resolutions, GWs with periods of ~20 minutes and above are detected.

Due to the resolution being used, the phase shift is determined from the 10-minute resolution time series data.

*Explanation of determination of Phase Shift*

The phase shifts were determined from the phases determined from each emission layer time series. The phase ($\phi$) is estimated from the Equation (2). The phase is given in hours. However, the phase needs to be estimated considering the start time of the data been used. This is to give the phase in relation to the start time of the time series. This is done by adding the individual dominant period obtained to the time series until it corresponds to the first hour that corresponds to the start time of the time series. For instance, the phase of the 04/12/204 OI GW event is 0.0744475 hours for GWs with a period of 0.424451 hours (~25.47 minutes). The period is added to the phase until 23.419253 (23:25:00 UT) was attained. A similar procedure was applied to the other emissions. In table 1, the result of the phases of the two waves selected are presented. From the phases obtained for the individual emission layers, the phase shifts can be estimated between each two consecutive layers as well as the first and the fourth layers.

**Table 1:** Estimated phase of 04/12/2004 GW event.

| Layers | $\phi_{\tau=25.24}$ | $\phi_{\tau=38.00}$ |
|---|---|---|
| OI | 23.419253 | 23.259999 |
| $O_2$ | 23.417403 | 23.312784 |
| Na D-Line | 23.829630 | 23.293561 |
| OH | 23.419026 | 23.683039 |

**Error Analysis**

As commented by the referee, error analysis has been performed in order to evaluate the errors associated with our calculations.

The error associated with each error layer has been performed to evaluate the impact on the result obtained. The error was assessed by estimating the standard error in the original data, the smoothed data and the harmonics. It is important to mention that the estimated standard error of the mean ($\sigma_M$) for OI 5577, O2, Na D-Line and OH intensities, temperature and time are presented in Table 2.

**Table 2:** Associated errors in the time series of the 04/12/2004 GW event.

| Layers | OI | $O_2$ | Na D-Line | OH |
|---|---|---|---|---|
| Original Data | | | | |
| $\sigma_{M_{Time}}(s)$ | ± 09.87097 | ± 09.87104 | ± 09.87093 | ± 09.87096 |
| $\sigma_{M_I}(R)$ | ± 07.33892 | ± 14.74540 | ± 04.07605 | ± 23.24270 |
| $\sigma_{M_T}(K)$ | | ± 01.72822 | | ± 03.13196 |
| Usable Data Range | | | | |
| $\sigma_{M_{Time}}(s)$ | ± 05.60431 | ± 05.60443 | ± 05.60471 | ± 05.60436 |

| | | | | |
|---|---|---|---|---|
| $\sigma_{M_I}$(R) | ± 06.95750 | ± 10.91080 | ± 01.05222 | ± 07.99789 |
| $\sigma_{M_T}$ (K) | | ± 00.67040 | | ± 00.41251 |
| Residual | | | | |
| $\sigma_{M_{Time}}$(s) | ±05.60432 | ± 05.60443 | ± 05.60471 | ± 05.60436 |
| $\sigma_{M_I}$(R) | ±00.96403 | ± 02.53697 | ± 00.20267 | ± 05.02443 |
| $\sigma_{M_T}$ (K) | | ± 00.23636 | | ± 00.25339 |

From Table 2, the estimated errors in the time, intensities and temperature are presented. In general, the errors associated with the original data is higher than that of the usable data and the residual. These values are less than the measurement errors of the intensities which is of the order of 5% whereas for intensities and 2-3K for O2 and 4-5K for OH (Wrasse et al., 2004).

The error associated with the fit was evaluated by estimating the cross correlation between the time series of the residual of the intensities and temperature and their respective harmonics are indicated in Figure 6a. The cross correlation of the intensities, that is, their residuals and harmonics ranged are 0.76, 0.79, 0.65 and 0.77 for IO, O2, NaD and OH intensities, respectively. For the temperature residuals and harmonics for O2 and OH are 0.52 and 0.67, respectively.

2. **In the processing section, "a Lomb-Scargle periodogram and Wavelet analysis were used to determine the dominant periods in the time series of each emission layer. At least a dominant peak is chosen and used to reconstruct new harmonics and over plotted on the residual." Lines 135-136.**

In this case, if there is any error in the harmonics chosen, how would this influence the result? The waves present appear to have a spectrum of associated periods. Is one chosen harmonic effectively characterizing the waves present?

**Response:**

In Table 2, the errors associated with the harmonics are presented. From these values, no significant errors were associated. Also, to assess the degree of deviation of the harmonics from the residual. As shown in the main text and in the response to item #1, except the intensity of the NaD, the remaining intensities have cross correlation above 0.75. This, however, shows that no significant differences exist between the residual and the harmonics. Therefore, there would be no significant influence on the results.

Yes, the one chosen characterized the wave event of interest. It is for this reason the reconstructed harmonics are over plotted over the residuals.

3. **Lines 146-147: "After the residual time series was determined, the periodicities were calculated. For these residuals, the dominant period are 25.47 min and 33.47 min."**

What process was used to determine these were the periods? Was a peak finding routine used with the PSD shown in figure 3d? Was this the period at all times? Was a particular time period chosen for each emission line? Are they all the same?

**Response:**

Lomb-Scargle periodogram and Wavelet analysis were used to determine the periods. Peak selection procedure considering the power spectral densities (PSD) were used to detect the significant periods. To determine whether the periods were present all time, the Wavelet analysis was implemented. As shown in Figure 3d (now 6d), it can be seen that the peak periods range between ~25 to 50 minutes and extend almost across the entire interval of the time series considered for all emission layers. In the case whether the periods are the same or not, we only considered the same period in each emission layer, or periods with deviations with ±15%. These periods are observed. The deviations were included in the choice of period because they are observed period. The result on the estimated intrinsic parameters are discussed later.

Furthermore, looking at figure 3a, the "reconstructed" signal appears to fit well for some portions of the night, but not for others. Was there a particular time used to determine the phase differences between the emission layers? Just a slight offset between the fit and actual data could result in significant differences for the vertical wavelength calculation.

**Response:**

As mentioned in the response to item #1, the start time of the time series are all the same. This is done to avoid offsets that could result in significant differences.

4. **Lines 152-153: "From the reconstructed time series, it is clear that all the emission layers are similar, indicating that the same GWs propagate through these layers."**

How was this determined? There are certainly similarities in all of the layers. This could also be expected for a ducted wave as well. It also appears that the wavelets in Figure 3d are not the same for all of the layers, so more explanation should be provided here. While it is mentioned that the wavelet shows the presence of waves from 30-90 minutes, there is still variability between the layers.

**Response:**

The reconstructed time series (harmonic) was constructed using equation (2), using the time of observation (x), amplitudes, observed period(s) of the GWs and phases of each emission layer. Similarities between layers indeed exist, however as mentioned, this could be due to ducted waves. As a result, the propagation conditions are investigated in the discussion section in order to verify whether or not they are ducted waves.

*Explanation on the variations of the periods in all layers*

In Figure 3d, even though similarities exist in the periodicities, some degree of variations can be seen. These variations can be attributed to the variations of the background wind, since the result obtained in the Lomb Scargle periodogram, and the Wavelet are observed periods.

**5. Section 3.4 needs more discussion about how the parameters were calculated, this is detailed in the following comments:**

-How was the Brunt Vaisala Frequency in Figure 4d calculated? What assumptions were made to calculate the potential temperature? How was dTpot/dz, the change in potential temperature with altitude calculated, and were there any assumptions made? Figure 4c shows a variable potential temperature, but the Brunt Vaisala frequency is plotted as a constant at each altitude.

**Response:**

The Brunt Väisälä frequency is estimated using Equation (5). During this wave event, SABER (Sounding of the Atmosphere using Broadband Emission Radiometry) instrument onboard the TIMED (Thermosphere Ionosphere Mesosphere Energetics Dynamics) satellite, made a passage ~735km away from the observation site. The temperature and pressure were used in the determination of the potential temperature, where $K/c_p = 0.286$. Using a first order derivative procedure in idl (interactive data language), the $d\theta/dz$, profile were determined. The potential temperature is the peak altitude of OH and $O_2$, which are ~87 and ~92 km where chosen. However, since, the temperatures of the OH and $O_2$ layers varies with time at fix altitude, so $d\theta/dz$ is estimated considering the value of $dz$ at the respective emission layers. The time series of $\theta$ is then estimated considering a constant $dz$.

After the estimation of Brunt Väisälä frequency from the profile determined from using SABER temperature, the Brunt Väisälä frequency at the respective emission layer peak altitude was chosen for the entire time. Due to your comment, the result of Figure 4c (now 7c) has been corrected. The profile has been estimated using the OH and $O_2$ temperature.

**6. What measured parameters are used for the energies? How is T'/T being calculated? Is the amplitude of the residual temperatures being used? How is the amplitude being calculated?**

**Response:**

The measured parameters used in the estimations of the energies are $k,l$, $m$, $k_H^2$, $\frac{T'}{T}$, and $N^4$. Basically, it can be said that except $g$, all other parameters are determined from observations. To determine the relative amplitude of the temperature time series, the $T'$ was first determined by subtracting the background from the from the temperature time series $\bar{T}$ (Narayanan et al., 2024). The amplitude of the gravity wave temperature perturbation was estimated by dividing the temperature perturbation by the background temperature. In the previous version of the manuscript (and mentioned in item #6), the vertical flux of the horizontal momentum was assumed to be estimated over the wave packet. Also, only the temperature fluctuations, $T'$, due to the gravity wave was used in the estimation of the potential energy.

However, due to your comments and questions, the entire calculations on the momentum flux and energies have been done all over again.

7. **Where are u' and v' being calculated from equation 4 for kinetic energy? What assumptions were made for the calculations of Ek shown in figure 4g?**

**Response:**

The $u'$ and $v'$ were calculated from the observed wind from meteor radar. No assumptions were made in the estimation of the $E_k$. Considering your comment regarding the temporal resolution of the meteor radar wind and the observed GW periods in item #12, the estimation of the $E_k$ for such waves have been removed.

8. **For horizontal momentum flux, as you show in equation 6 (now 7), this is typically assumed to be the average vertical flux of horizontal momentum over a wave packet. Over what time period is the wave packet defined? What is being used at the T' calculation?**

**Response:**

As mentioned earlier, it is the average vertical flux of horizontal momentum over a wave packet, thus, no period over the time series was defined. Here, the estimated $T'$was used.

9. **For equations 6/7, on line 175 it says "where rho_o is the density at the emission layers" but it is not clear where or how this density was obtained to make the calculation shown in Figure 4e.**

**Response:**

The density, $\rho_0$, used in equations 6 and 7 was obtained from SABER sounding close to the observation site during each GW event. The density information has been included in the text.

10. **For equation 6/7, how was kh, the horizontal wavenumber, calculated from the data? This was not discussed elsewhere.**

**Response:**

The horizontal wavenumber, $k_H$, was estimated from the horizontal wavelength estimated in the Equation A5 in Appendix A2, using the relation

$$k_H = \frac{2\pi}{\lambda_H},$$

where, $\lambda_H$ is the horizontal wavelength. This has been defined in the main text.

**11. Similarly, how was the intrinsic frequency calculated or measured for the MF calculation?**

**Response:**

The intrinsic frequency, $\omega$, can be estimated from the expression, $\omega = \omega_0 - kU - lV$, where $\omega_0 = 2\pi/\tau=$. $U$ and $V$ are the zonal and meridional wind speed, respectively, at each peak emission altitude. This has been defined in the main text.

**12. Line 187-188: "Since the meteor radar wind has a temporal resolution of one (1) hour, Ek at each hour was determined and presented in a contour plot"**

The meteor radar is giving you the background wind. However, with a 1 hour resolution, the meteor radar is unable to give the perturbations u' and v' associated with the gravity wave that would be necessary to calculate Ek. The gravity wave periods present are all less than 1 hour. These Ek calculations are not correct.

**Response:**

Well, noted. As a result, the estimation of $E_k$ and it discussion has been removed. This aspect will be left for a companion paper.

**13. Lines 193-194: "The spectral analysis technique described in Wrasse et al 2024 was used to determine the horizontal wavelength, period, phase speed, and propagation direction"**

More details need to be given here about what exactly was done. This is a very cursory explanation for a significant calculation within the manuscript.

In the referenced 2024 paper, it appears that Fig 1 is a flow chart and Figure 2 shows individual images and keograms. This case presented there is a little different because the waves were very clear both in the individual images and in the keograms. In the manuscript here, and the data presented in A2, the waves are not necessarily clear in the keogram. It is also a little strange that individual images are not shown. Looking at data in Tabe 1, the determined kh was between 20-35km, which should be within the field of view of the imager. Why were spatial images not included? The horizontal wavelengths can easily be obtained from the images themselves and not a keogram. There needs to be more discussion about how all of these parameters were obtained.

**Response:**

Thanks for your comments! Based on your comments, the means by which the wave parameters are estimated are discussed below. This discussion has been included in Appendix A1.

In order to extract the parameters of gravity waves, a discrete Fourier transform based spectral analysis was used. First, a region containing GW oscillations were selected in both zonal and meridional component of the keogram components, as shown in Figure **xy**. Note that the same area in each of the components were considered for analysis. Next, a discrete Fourier transform (equation **A1**) is applied to the selected areas.

$$F(\omega) = \sum_{n=0}^{N-1} f(t)e^{\frac{-2\pi\omega n_i}{N}}$$

in which $F(\omega)$ is the transform of the Fourier function $f(t)$, $\omega = 0, \ldots, N-1$ is the frequency index, and $N$ is the number of points in time series within the selected regions. Then, the cross spectrum defined by

$$C(x) = F_s(\omega)F_{s+1}^*(\omega),$$

in which $C(\omega)$ is the cross spectrum between two time series and $F_s(\omega)$ and $F_{s+1}^*(\omega)$ represent the Fourier transform of the series $f_S(t)$ and $f_{s+1}(t)$, respectively. $F_{s+1}^*(\omega)$ is the complex conjugate of $F_{s+1}(\omega)$. The one-dimensional cross power spectrum is defined by the quadratic modulus, |C2|.

The amplitude of the cross power spectrum is then determined using $\sqrt[2]{|C^2|}$, with the phase of the cross spectrum defined by

$$\Delta\psi = tan^{-1}\left\{\frac{Im(C(\omega))}{Re(C(\omega))}\right\}, -\pi \le \psi \ge \pi.$$

The phase difference between these time series caused by the wave propagation is considered to be the frequency $\omega$, corresponding to the maximum amplitude. From the above estimations, the wave parameters are determined as follows:

Period (min):

$$\tau = \frac{1}{|f(\omega)|};$$

Horizontal wavelength (km):

$$\lambda_H = \frac{\lambda_{NS}\lambda_{EW}}{\sqrt{\lambda_{NS} + \lambda_{EW}}},$$

Where, wavelength (km) for the zonal and meridional components $(\lambda_{NS}, \lambda_{EW})$ is $\lambda_{NS,EW} = \frac{\Delta d}{\Delta\psi/360°}$, in which $\Delta d$ is the distance between the time series.

The horizontal phase velocity $C_H$(m/s), and phase propagation direction $\phi$(°), are determined by

$$C_H = \frac{\lambda_H}{\tau} \ and \ \phi = cos^{-1}\left(\frac{\lambda_H}{\lambda_{NS}}\right)$$

**14. Lines 200-207: For event 1 "the dominant periods used in the reconstruction of the waves of these events are 00.42 hr (25.47 min) for all emission layers, and 00.50 hr (30.29 mins) for IO 557.7, O2 (0-1) and NaD. However, the period of the OH (6 - 2) was 0.55 hr (33.47min)."**

This is a bit contradictory to read. It sounds like there were three different periods used here. Why are there so many different periods? Shouldn't they all be the same? These are also very close periods, close enough that a 2-minute measurement resolution would suggest a slight error in the fit could describe differences in presumed periods. Can it be demonstrated that this is not a fitting error and these periods are real? From Figure 5b, it looks like there is a lot of variability in periods over the dataset.

**Response:**

These lines have been re-written for clarity as below and in the main text. Also, regarding obtaining the same observe period in all emission layers depends on the background wind variations. Meteor radar wind showed some variations with altitude, and this can change in the observed period. In order to demonstrate that the obtained periods are real, the Lomb Scargle periodogram and the Wavelet analysis was applied to the original data to see the periods present before the application of the fit.

**15. Figure 5a makes it appear that there is little to no phase change between the different layers/altitudes. This is usually indicative of a ducted wave. Yet, the vertical wavelength listed in Table 1 is only 10km. Wouldn't there would be more variation in phase over the layers if the vertical wavelength were only 10km?**

**Response:**

That is right. In Figure 5a (now 7a) there is little to no phase change, especially at the beginning of the data used. As pointed out that this kind is an indicative of ducted waves, background studies has been conduct to investigate the propagation characteristics of this waves within the altitude range of the emission layers.

The vertical wavelength had been checked and verified, but the estimation still yielded 10 km.

**16. Lines 210-215: descriptions of the wavelet plots are given, but it really is not clear how the peaks/wave periods were determined. The plots show a broad spectrum. How was one particular peak chosen to represent a wave across the entire dataset?**

**Response:**

In the current version of the manuscript, the procedure of determination of the peaks had been described.

Two periods were chosen for the reconstruction. For the GW event of 25 May 2006, the wavelet analysis showed a period (of 70 mins) with strong amplitude was observed in Figure 5d(i), thus for the OI emission layer. However, this period only appeared in the OH wavelet (in Figure 5d(vi). Because of this, only two periods were observed and all these two can be seen with a peak ~30 minutes, which extend to about ~50 minutes. For the case of event of 04 December 2004, the period are quite consistent in all the four emission layers. As mentioned earlier, the procedure of the period selection has been discussed in Section 4 (Result).

**17. Table 1: Are there any errors associated with these measurements/calculations?**

**Response:**

Yes. There are errors associated with the measurements and calculations. These errors are included in the current version of the manuscript.

**18. Table 1: It is still not clear how the vertical wavelength was calculated from the measurements.**

**Response:**

In the processing section (i.e., 3.3), the procedure to the estimation of the vertical wavelength has been elaborated in detail.

**19. Lines 218-291: "Only the potential energy for Event #02 could be determined due to unavailability of observed winds. Hence, no estimated values for kinetic energy and subsequently total energy were presented in Table 1."**

It would appear based on what has been presented in this paper that Ek cannot be calculated for any of the events.

**Response:**

This will exactly be the case. As mentioned in your previous comment in in the previous comments regarding the estimation of the Ek, the temporal resolution of the meteor radar wind will not capture the spectrum of GWs under study. Due to this, the estimation and discussion of the kinetic energies have been removed.

**20. Lines 220-225 and Figure 5c and d: The wavelet analysis shows periods that are all over the place. The final fitted waves based on "dominant periods" in the wavelet are shown in Figure 5c. However, the original data plotted with the fit are never shown like they were for event 1 in Figure 3. This needs to be included.**

**Response:**

Thanks for the comment. The original data has been plotted with fit in the current version of the manuscript.

**21. Lines 237-238: "two events with similar periods were selected. For Event #01, two dominant periods were detected, however, the first period present no phase change, implying it is possibly a ducted wave."**

It is still not clear how the "dominant periods" were chosen. It would also appear that there is little to no phase change between the different layers.

**Response:**

In item 15, a response regarding the investigation of the propagation characteristics of GW event of 04 December 2004 if the waves during this event are ducted or not. A follow up discussion on the potential and momentum flux was made to investigate the dynamics of the momentum flux in such conditions. The selection of the dominant period was based on the PSD using a procedure in IDL.

**22. Line 229: "For Event #02, the two dominant periods are within the gravity wave spectrum."**

This is not at all clear from the wavelet.

**Response:**

Event two has been re-analyzed and plot. However, further analysis showed that this case cannot be used, hence another case was selected.

23. 23.Line 233-235: **"From the phases of the GWs of Event #01, OH leads NaD by 08.60 min, whereas NaD leads O2 by 01.21 min. O2 lags OI by 03.25 min. A consistent phase lead can be observed from OH through NaD to O2 except between O2 and OI, where a phase lag is observed."**

Where was this demonstrated in the data? Figure 5a shows nearly identical phases over each layer. Furthermore, the resolution of the measurements would not allow for these sorts of phase differences to be measured. 1.21 minutes is less than the resolution of the measurement.

**Response:**

It is true that the resolution of the measurement will not permit the determination of a phase less than 2 minutes. However, considering the altitude difference between the two emission layers and if it is the same wave propagating through the two layers with a high velocity, it is possible there will be near no phase difference. We can also consider the possibility that the overlapping of the two emission layers might possibly contribute as well as reflection.

24. Line 235: *"The phase lag observed between the emission layers of O2 and OI was induced by the background wind due to a shear."*

Where? How was this proven mathematically in any of the previous data / measurements / calculations presented?

**Response:**

Phase change can possibly occur when wave reflection occurs, and wave reflection are induced by wind. This implies possible formation of regions of $m^2 < 0$. The characteristics of ducts are observed during the two events considered in the may possible cause this kind of behavior.

25. Line 236-238: *"Despite this phase lag, the mean phase propagation of these GWs shows that OH leads OI by ~06.58 min. Using this phase information and the period, Figure 6(a) is produced. Clearly, it is observed that the similar GW oscillation in the OH (red line) emission layer leads to the OI (green line) emission."*

None of these phase differences have clearly been shown. The calculations to obtain them have not been clearly demonstrated.

A sinusoid can be fit to anything. There needs to be a determination of how good the fit actually is. Figure 6 shows "reconstructed waves" but it is not clear where this is obtained from. Is this from the data shown in Figure 3? The original data should be plotted with the fits.

The discussion of "leading" and "lagging waves" needs to be tied back to the data more clearly, and this also needs to be put into context of the actual resolution of the measurements. Ultimately, instrument resolution is going limit the ability to determine phase differences.

Response:

Thanks very much for the comments and suggestions. They items mentioned above have been addressed in the new version of the manuscript

**26. The section "Momentum Flux and Wave Energy" will likely need to be redone with more explanation regarding how the different parameters were calculated and what assumptions were made for the calculations. Most importantly, how was the average temperature perturbation determined for the wave packets present in the measurements?**

Additionally, it does not seem that the kinetic energy calculations for a wave are not correct.

**Response:**

Due to your earlier comments, the calculation of the kinetic energy was removed. For the momentum flux estimation, more clarification has been given as to how the individual parameters have been determined.

**27. The conclusions that there are upward and downward propagating gravity waves need to be better supported. It seems like arbitrary wave periods were chosen from the wavelet analysis, and sinusoids were plotted based off of this. There needs to be more quantitative analysis performed and a justification for the reconstructed waves provided.**

**Response:**

As mentioned earlier, the wave periods were not chosen arbitrarily. They were chosen based on the intensity of the PSD, thus, the peak PSD corresponding to the time and period (for Wavelet analysis) and for the Lomb-Scargle (the period corresponding to the peak PSD).

---

## Referee Report (RR1)

General review:

This paper investigates the characteristics of vertical propagating gravity waves (GWs) using observational data from co-located photometers, all-sky imagers, and meteor radars at São João do Cariri. The focus is on quantifying the momentum flux and potential energy associated with GWs as they propagate vertically through various atmospheric layers. The study leverages airglow emissions and rotational temperature data to analyze phase progression and vertical wavelength across different altitudes. The results emphasize the dynamics of momentum and energy transfer under varying propagation conditions, including ducted and near-vertical propagation scenarios.

Areas for improvement:

The paper includes a vast amount of technical detail but lacks an initial high-level summary that could provide readers with an overview of the key findings and their significance. Add a summary at the beginning that outlines the study's objectives, methodology, key findings, and implications in a concise format.

The study does not explore how the findings can be applied to broader atmospheric modeling or real-world scenarios like weather prediction or satellite operation. Please Add a summary at the beginning that outlines the study's objectives, methodology, key findings, and implications in a concise format.

The selection of the two gravity wave events could benefit from a more explicit justification regarding their uniqueness or representativeness. Please Provide a stronger rationale for selecting the specific GW events analyzed and discuss how they compare to other observed events.

Minor suggestions:

Line 567, thee -> the

---

## Author Response (AR2)

**Author's Response**

The authors would like to appreciate the reviewers for their comments and suggestions.

We have addressed and responded to the comments and suggestions of the reviewers accordingly. The text highlighted in blue are the responses. These responses are implemented in the manuscript and also highlighted in blue as well.

**General review:**
This paper investigates the characteristics of vertical propagating gravity waves (GWs) using observational data from co-located photometers, all-sky imagers, and meteor radars at São João do Cariri. The focus is on quantifying the momentum flux and potential energy associated with GWs as they propagate vertically through various atmospheric layers. The study leverages airglow emissions and rotational temperature data to analyse phase progression and vertical wavelength across different altitudes. The results emphasize the dynamics of momentum and energy transfer under varying propagation conditions, including ducted and near-vertical propagation scenarios.

**Areas for improvement:**
The paper includes a vast amount of technical detail but lacks an initial high-level summary that could provide readers with an overview of the key findings and their significance. Add a summary at the beginning that outlines the study's objectives, methodology, key findings, and implications in a concise format.

**Response:**

Momentum flux and propagation dynamics of two vertical propagating atmospheric gravity waves (GW) are studied using observation at São João do Cariri, Brazil (36.31°W; 07.40°S) by co-located photometer, all-sky imager, and meteor radar. Time series of atomic oxygen green line (OI 557.7 nm), molecular oxygen ($O_2$), sodium D-line (NaD), and hydroxyl (OH) airglow intensity variations measured by the photometer were used to investigate the vertical characteristics and phase progression of the GWs with similar/same period across these emission layers. The horizontal parameters of the same GWs were determined from the OH airglow images, whereas the intrinsic parameters of the horizontal and vertical components of the GWs were estimated with the aid of the observed winds. Using the phase of the GWs at each emission layer, the characteristics of the phase progression exhibited near vertical propagations under a duct background propagation condition. This indicates that the duct contributes significantly to the observed near vertical phase propagation. The GW momentum flux and potential energy were estimated using the rotational temperatures of OH and O2, revealing that the time series of momentum fluxes and potential energies are higher in the $O_2$ emission than OH, a transfer of momentum and energy across OH to the $O_2$ altitude. These results reveal the effect of a duct on vertical propagating GW and associated momentum flux and potential energy transfer from the lower to the upper altitudes in the mesosphere.

The study does not explore how the findings can be applied to broader atmospheric modeling or real-world scenarios like weather prediction or satellite operation. Please Add a summary at the beginning that outlines the study's objectives, methodology, key findings, and implications in a concise format.

**Response:**

Numerous studies (e.g., Fritts et al., 2006; Suzuki et al., 2013; Love and Murphy, 2016 Kaifler et al., 2020) employed some of these observational techniques to explore the subject of dynamics of GWs and their momentum fluxes and potential energies. Fritts et al. (2006) investigated the momentum fluxes due to GW activities in the MLT region using wind measurement from incoherent scatter radar (ISR) at Arecibo observatory. Using a time resolution of ~50 minutes, between 71 and 95 km, they quantified GW momentum fluxes profiles. VHF mesosphere-stratosphere-troposphere (MST) radar measurements situated near Davis station (68.5°S, 78.0°E) were used by Love and Murphy, 2016 to study the hourly averaged profiles of GW momentum fluxes between 79 and 90 km along the day. Love and Murphy, (2016g) investigated hourly averages of momentum fluxes of days considered within the period of 14 December 2014 to 6 January 2015 as well as GW intermittency with altitude. Using a co-located observation, Suzuki et al. (2013) investigated the vertical propagation of GW from the lower to the upper atmosphere at the Arctic Lidar Observatory for Middle Atmosphere Research (ALOMAR) station (69.31°N, 16.01°E). Using the sodium (Na) airglow imager, the horizontal structure of GWs is observed, whereas the ALOMAR Rayleigh/Mie/Raman (RMR) lidar and sodium lidar reveals the two-dimensional vertical structure of GWs between the stratosphere and the lower thermosphere. This coincident observation permitted the study of horizontal and vertical characteristics of GWs and the momentum flux at the Na airglow altitude. Kaifler et al. (2020), on the other hand, used high temporal and vertical resolution Lidar to study the derived time series absolute momentum fluxes of mountain waves at 40 km and profile of mean and peak of the absolute momentum fluxes between 10 and 80 km.

The above-mentioned works have by far contributed to quantifying the characteristics of the momentum fluxes of GWs and mountain waves (MWs) through statistical and case studies. However, none of these studies explored the aspect of how the momentum fluxes and possibly potential energies would behave under different vertical and horizontal propagation of GWs. Therefore, the standing question this work addresses is, the behaviour of the momentum flux and potential energy of GW under different phase/energy propagations, thus, upward, downward or ducted. Using the vertical phase propagations of GWs, the energy propagation can be determined.

For this, investigation was made on the vertical characteristics of GWs with similar or the same period propagating vertically across four (4) airglow emission layers: atomic oxygen green line (OI 557.7nm), molecular oxygen ($O_2$ - 864.5nm), sodium D-line (NaD-589.0nm), and hydroxyl (OH) (6-2) band. Next is the determination of the phases of the wave at each layer and, consequently, the phase propagation. The horizontal characteristics of the same GWs are estimated from OH all-sky images. Using observed wind, the intrinsic parameters were also estimated. Having determined and classified these GWs as vertically propagating, the background propagation conditions are studied as well as the potential energy and momentum flux at the OH and $O_2$. The temperature measurements employed to determine the potential energy and momentum flux were obtained using the rotational temperature at the OH and $O_2$ emission layers. The dynamics

of the GWs potential energy and momentum flux under the determined propagation condition were then studied. It was discovered that the vertical propagation of the cases selected were controlled by the background conditions imposed by the wind and temperature.

The selection of the two gravity wave events could benefit from a more explicit justification regarding their uniqueness or representativeness. Please Provide a stronger rationale for selecting the specific GW events analysed and discuss how they compare to other observed events.

**Response:**

To select the cases, they must satisfy the following criteria:

1. GWs with similar or nearly the same periods must be observed in all the four (4) emission layers;
2. similar or nearly the same periods observed in the four airglow intensity variations must be present in the OH and O2 rotational temperature;
3. similar or the same periods observed in item #01 must be present in the OH and possibly $O_2$ all-sky images;
4. the periods of the GWs at each emission layer must be present in the time series for 3 hours or more.

Besides selecting these two GW events due to the presence of nearly the same periods in the four emission layers, these events permit the exploration of the dynamics of GWs using (1) observed variables in the MLT region at high temporal resolution and (2) derived momentum flux.

To achieve this, all three (3) criteria must be satisfied. Most of the observed cases that met criteria #3 either lack criteria #1 or #2. This is to say that most of the cases have nearly the same period in three emission layers and one rotational temperature, like the work of Nyassor et al., 2018. On the other hand, majority of the events have similar or the same period in just two (2) emission layers.

**Minor suggestions:**

Line 567, thee -> the (now Line 627)

**Response:** the word "thee" has been changed to "the".